# Perception Stitching: Zero-Shot Perception Encoder Transfer for Visuomotor Robot Policies

Pingcheng Jian[1]  Easop Lee[1]  Zachary Bell[2]  Michael M. Zavlanos[1]  Boyuan Chen[1]
[1]Duke University  [2]Air Force Research Laboratory
generalroboticslab.com/PerceptionStitching

**Reviewed on OpenReview:** `https://openreview.net/forum?id=tYxRyNT0TC`

## Abstract

Vision-based imitation learning has shown promising capabilities of endowing robots with various motion skills given visual observation. However, current visuomotor policies fail to adapt to drastic changes in their visual observations. We present Perception Stitching that enables strong zero-shot adaptation to large visual changes by directly stitching novel combinations of visual encoders. Our key idea is to enforce modularity of visual encoders by aligning the latent visual features among different visuomotor policies. Our method disentangles the perceptual knowledge with the downstream motion skills and allows the reuse of the visual encoders by directly stitching them to a policy network trained with partially different visual conditions. We evaluate our method in various simulated and real-world manipulation tasks. While baseline methods failed at all attempts, our method could achieve zero-shot success in real-world visuomotor tasks. Our quantitative and qualitative analysis of the learned features of the policy network provides more insights into the high performance of our proposed method.

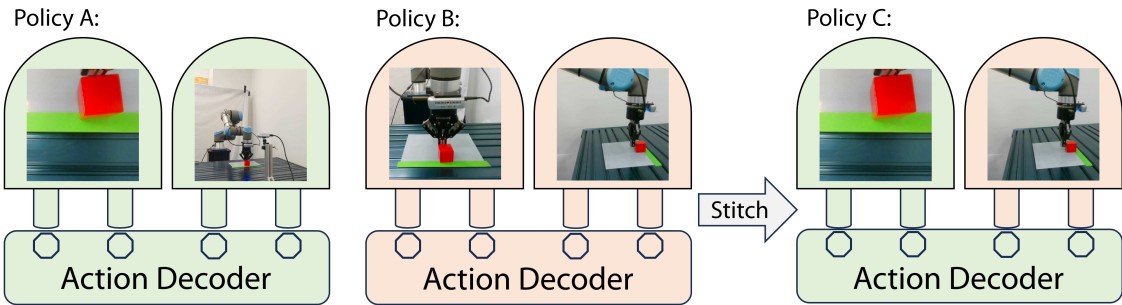

Figure 1: **Perception Stitching**: "Policy A" was trained with an in-hand camera and a front-view camera. "Policy B" was trained with a close-up camera and a side-view camera. Perception Stitching enables zero-shot stitching of the original Policy A and B by reusing their relevant components for each sensing configuration to form a "Policy C". "Policy C" can maintain strong zero-shot transfer performance with an in-hand camera and a side-view camera.

## 1 Introduction

Despite recent advances in vision-based imitation learning for acquiring diverse motor skills (Chi et al., 2023; Fu et al., 2024), a significant challenge in deploying visuomotor policies in real-world settings is ensuring the perceptual configurations are identical during training and policy execution. With the growth of hybrid robot datasets where robots are trained to perform similar tasks across the world, it remains difficult to share their learned experiences, even under the same task but with different visual observations. Such a challenge often stems from the unique configurations and perspectives each camera setup brings, which, while enriching the dataset, complicates the sharing of learned experiences across different systems. Different institutions worldwide usually place different sensors at different perspectives when collecting their datasets. In these

cases, previously collected datasets or trained policies are not interchangeable, and new training data must be collected in each environment.

Instead, what if we could directly stitch the perception encoder trained in one visuomotor policy to the rest of the components of another visuomotor policy? This approach would enable zero-shot transfer of the trained visuomotor policies to a novel combination of perceptual configurations. To address the challenges associated with zero-shot transfer across different settings, previous efforts have aimed to learn an invariant feature space or universal representation for quick adaptation to new environments, with approaches ranging from extensive pre-training with video data (Nair et al., 2022), employing contrastive learning between two policies to find a common feature space (Gupta et al., 2017), and concentrating on low-dimensional data over vision-based observations (Jian et al., 2023). In particular, recent studies have proposed to achieve fast policy transfer (Jian et al., 2023) by aligning the latent representations of different perception encoders with relative representation (Moschella et al., 2022). However, it remains unclear how to scale similar approaches from few-shot transfer to zero-shot transfer and high-dimensional observations.

We present **Perception Stitching (PeS)** (Fig. 1) to enable zero-shot perception encoder transfer for visuomotor robot policies. PeS advances the previous studies through a novel training scheme under various camera configurations and effectively processing high-dimensional image data. Our approach can train modular perception encoders for specific visual configurations (e.g. camera parameters and positions) and reuse the trained perception encoders in a novel environment in a plug-and-go manner. In the simulation, we evaluate PeS in five different robotic manipulation tasks, each with seven unique visual configurations. It constantly shows significant performance improvement compared to four baseline methods and two ablation studies. We also evaluate PeS in four real-world manipulation tasks. While the baseline struggles to get any successful attempts, PeS achieves pronounced success rates, indicating that directly stitching modular perception encoders in the real world has been turned from impossible to possible by this work. Additionally, we contribute quantitative and qualitative analysis to provide more insights on the high performance of our proposed method.

## 2 Related Work

**Learning Visuomotor Policy for Robotic Manipulation** A wide range of previous work has focused on learning visuomotor policy for robotic manipulation (Levine et al., 2016; Finn et al., 2016; 2017b; Kalashnikov et al., 2018; Srinivas et al., 2018; Ebert et al., 2018; Zhu et al., 2018; Rafailov et al., 2021; Jain et al., 2019; Hämäläinen et al., 2019; Florence et al., 2022; Brohan et al., 2022; 2023; Padalkar et al., 2023; Sermanet et al., 2018). Certain works have investigated the impact of camera placements (Zaky et al., 2020; Hsu et al., 2022), design of the hardware and software (Zhao et al., 2023; Fu et al., 2024; Kim et al., 2023), novel network architectures and optimization techniques (Dasari & Gupta, 2021; Kim et al., 2021; Zhu et al., 2023; Abolghasemi et al., 2019; Ramachandruni et al., 2020; Brohan et al., 2023; 2022; Padalkar et al., 2023; Chi et al., 2023; Li et al., 2023). In this work, we show that perception encoders trained with Behavior Cloning (BC) (Pomerleau, 1988) often lack the flexibility for module reuse. Our Perception Stitching (PeS) method enables perceptual knowledge reuse and facilitates zero-shot transfer between diverse visual configurations, advancing existing research on fast adaptable visuomotor policy design.

**Robot Transfer Learning** Transfer learning has long been considered a primary challenge in robotics (Tan et al., 2018; Taylor & Stone, 2009). In the context of reinforcement learning, many previous work transfer different components such as policies (Devin et al., 2017; Konidaris & Barto, 2007; Fernández & Veloso, 2006), parameters (Finn et al., 2017a; Killian et al., 2017; Doshi-Velez & Konidaris, 2016), features (Barreto et al., 2017; Gupta et al., 2017), experience samples (Lazaric et al., 2008), value functions (Liu et al., 2021; Zhang & Zavlanos, 2020; Tirinzoni et al., 2018), and reward functions (Konidaris & Barto, 2006). In imitation learning, additional studies have made progress via domain adaptation (Kim et al., 2020; Yu et al., 2018), querying unlabeled datasets (Du et al., 2023), abstracting and transferring concepts (Lázaro-Gredilla et al., 2019; Shao et al., 2021), or conditioning on other information such as language instructions (Stepputtis et al., 2020; Lynch & Sermanet, 2020; Jang et al., 2022) and goal images (Pathak et al., 2018). Our work focuses on neural network policy sub-module reuse through direct stitching to achieve zero-shot transfer.

Sim-to-real transfer is another important topic within transfer learning (Sadeghi et al., 2018; James et al., 2019; Zhang et al., 2019; Tobin et al., 2018; Mehta et al., 2020; James et al., 2017; Nguyen et al., 2018; Rusu

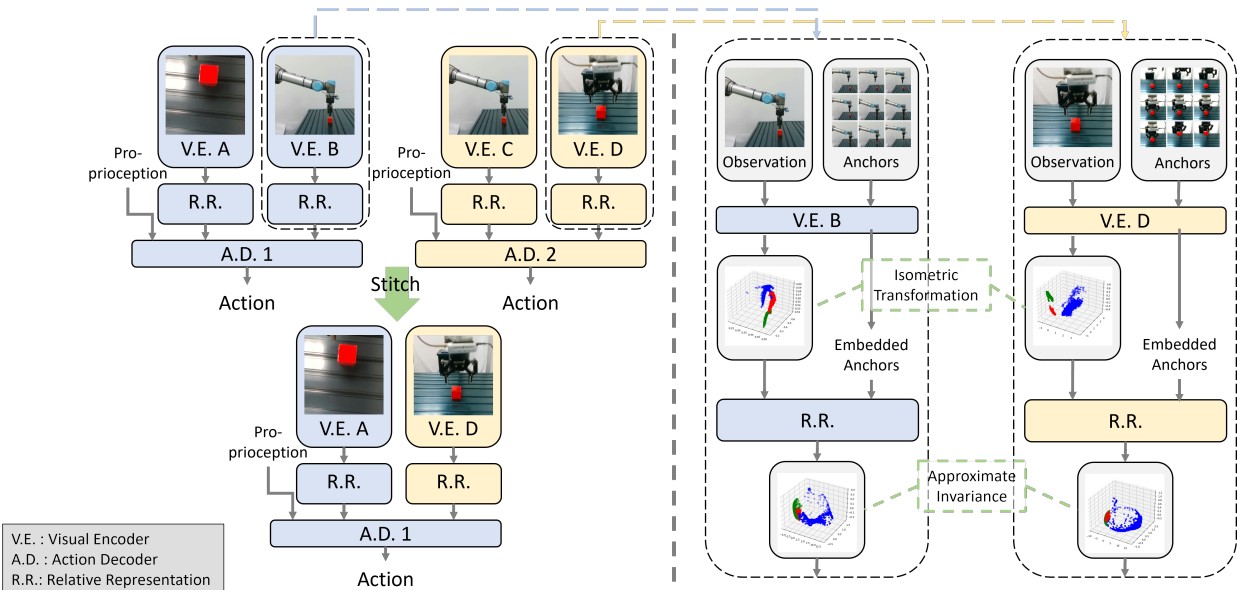

Figure 2: **Method Overview.** Two visual encoders process the RGB images from two cameras separately, and the latent representations are concatenated with the proprioception of the robot end effector state. The original latent representations of the images are observed to have an approximate isometric transformation relationship. Relative representations with disentanglement regularization can maintain an approximate invariance and, therefore, help achieve high zero-shot transfer performance.

et al., 2017). To merge the sim2real gap, previous methods include domain adaptation (Zhang et al., 2019; Tobin et al., 2018; Mehta et al., 2020; James et al., 2017), adopting a progressive network (Rusu et al., 2017; 2016), fine-tuning the visual layers (Sadeghi et al., 2018), training a generator that translates real-world images to canonical simulation images (James et al., 2019), or training a CycleGAN (Zhu et al., 2017) to synthesize real images from simulation images(Nguyen et al., 2018). Most recent research has proposed to learn invariant or universal feature representations for transfer learning (Gupta et al., 2017; Jian et al., 2023; Nair et al., 2022). Our work fits in this category. Unlike previous studies, our method does not require a large amount of online data for pre-training (Nair et al., 2022) or training two policies simultaneously(Gupta et al., 2017). Compared with a similar previous work by Jain et al. (2019), our method is not limited to low dimensional observations, does not require few-shot fine-tuning, and can solve much more difficult tasks.

**Compositional Robot Learning** Compositional robot learning reuses the learned knowledge saved in some portion of the policy network in new tasks instead of training from scratch (Pfeiffer et al., 2023; Devin, 2020; Alet et al., 2018; Chen et al., 2020). It has achieved promising results in multi-task learning (Yang et al., 2020; Hussing et al., 2023; Chen et al., 2022; Kwiatkowski et al., 2022; Hu et al., 2022), transfer learning (Devin et al., 2017; Jian et al., 2023; Gupta et al., 2017), and lifelong learning (Mendez et al., 2022; Mendez & Eaton, 2022; Méndez, 2022). Some previous works train a graph of network modules and generate different paths to connect specific modules for different tasks (Yang et al., 2020; Mendez et al., 2022), but the network modules cannot be reused out of the graph it embeds in. Devin et al. (2017) reuses the network module without aligning latent space representations and thus fails to achieve satisfying performance in more complex tasks. Jian et al. (2023) aligns the latent spaces by selecting anchor states and calculating relative representations (Moschella et al., 2022) at the module interface, which no longer requires simultaneously training multiple policies. However, the results are limited to low-dimensional observations and few-shot fine-tuning in simple tasks such as pushing. In this work, we extend its success to high-dimensional image observations and more difficult tasks such as cube stacking and door opening with zero-shot transfer.

# 3 Perception Stitching: Learning Reusable Perception Network Module

Perception Stitching (PeS) (Fig. 2) is a compositional vision-based robot learning framework that allows zero-shot transfer of perceptual knowledge between different camera configurations. It is designed to be compatible with conventional behavior cloning algorithms (Bain & Sammut, 1995; Ross et al., 2011; Torabi et al., 2018; Chi et al., 2023), various visual encoder structures such as CNN (LeCun et al., 1989) and ResNet

(He et al., 2016), and action decoder structures such as MLP (Hinton et al., 2012; Mandlekar et al., 2021) and LSTM (Hochreiter & Schmidhuber, 1997). In this section, we will demonstrate the two main components of PeS: the modular visuomotor policy design and the latent space alignment for transferable representation.

## 3.1 Modular Visuomotor Policy Design

Consider an environment $E$ with observation $o^E$. We denote the observations from two camera views as $o_1^E$ and $o_2^E$, and the robot proprioception of the end effector position, orientation, and gripper open width as $o_p^E$. For an MLP-based policy, we denote it as $\pi_E(a^E \mid o^E)$ parameterized by a function $\phi_E(o^E)$. For an RNN-based policy, we denote it as $\pi_E(a^E, h_{t+1}^E, c_{t+1}^E \mid o^E, h_t^E, c_t^E)$ parameterized by a function $\phi_E(o^E, h_t^E, c_t^E)$, where $h_t^E$ is the hidden state of the RNN network at time step $t$ and $c_t^E$ is the cell state. The visuomotor policy can be decomposed into the visual encoder $g_1^E$ for camera 1, the visual encoder $g_2^E$ for camera 2, and the action decoder $f^E$. We can represent the MLP-based policy as Eq. 1 and the RNN-based policy as Eq. 2.

$$\phi_E(o^E) = \phi_E(o_1^E, o_2^E, o_p^E) = f^E(g_1^E(o_1^E), g_2^E(o_2^E), o_p^E), \tag{1}$$

$$\phi_E(o^E, h_t^E, c_t^E) = \phi_E(o_1^E, o_2^E, o_p^E, h_t^E, c_t^E) = f^E(g_1^E(o_1^E), g_2^E(o_2^E), o_p^E, h_t^E, c_t^E), \tag{2}$$

Without loss of generality, we demonstrate the perception stitching process with the MLP-based policy. With two visuomotor policies $f^{E_1}(g_1^{E_1}(o_1^{E_1}), g_2^{E_1}(o_2^{E_1}), o_p^{E_1})$ and $f^{E_2}(g_1^{E_2}(o_1^{E_2}), g_2^{E_2}(o_2^{E_2}), o_p^{E_2})$ in two environments $E_1$ and $E_2$ with different cameras, we define perception stitching as constructing another visuomotor policy network $f^{E_1}(g_1^{E_1}(o_1^{E_3}), g_2^{E_2}(o_2^{E_3}), o_p^{E_3})$ by initializing the visual encoder 1 with parameters from $g_1^{E_1}$, visual encoder 2 with parameters from $g_2^{E_2}$, and action decoder with parameters from $f^{E_1}$. Then this stitched policy is zero-shot transferred to the new environment $E_3$ with the same perception configuration 1 as that of $E_1$ and perception configuration 2 as that of $E_2$. For example, as shown in Fig. 2 (left half), we train policy $\phi_{E_1}$ in $E_1$ with an in-hand camera and a side-view camera, and policy $\phi_{E_2}$ in $E_2$ with a side-view camera and a front view camera. Now, if we need a policy for $E_3$ with an in-hand camera and a front-view camera, we stitch the visual encoder $g_2^{E_2}$ to the $g_1^{E_1}$ and $f^{E_1}$ to form a stitched policy that directly works in $E_3$. Note that though we formalize PeS set up with dual-camera settings, PeS is not constrained with only two cameras, as demonstrated in our experiments in the Section 4.2. In addition, although we choose the action decoder 1 for the experiments in section 4, the choice of which action decoder to be used doesn't affect the performance of PeS, and the experiment results of comparing the influence of the two action decoders are presented in Appendix D.

## 3.2 Latent Space Alignment for Transferable Representation

**Relative Representation** Simply stitching one portion of a neural network to another neural network usually cannot yield optimal performance in the target environment due to the misalignment of latent space (Jian et al., 2023; Devin et al., 2017). Previous works have observed an approximate isometric transformation relationship between the latent representations trained with different random seeds (Moschella et al., 2022) and different robot kinematics (Jian et al., 2023). These isometric transformations include rotation, reflecting, rescaling, and translation. In this work, we observe a similar phenomenon in the latent spaces of the visual encoders. Hence, we calculate a relative representation at the latent space (Jian et al., 2023; Moschella et al., 2022) to align the latent features from different policy modules.

As shown in Fig. 2, we first collect a set $\mathbb{A}$ of anchor images $\boldsymbol{a}^{(j)}$ from the dataset $\mathbb{D}$ for the behavior cloning: $\mathbb{A} = \{\boldsymbol{a}^{(j)}\} \subseteq \mathbb{D}$. By applying the visual encoder $g$ to both the input image $\boldsymbol{s}^{(i)}$ ($\boldsymbol{s}^{(i)} \in \mathbb{S}$) and the anchor image $\boldsymbol{a}^{(j)}$, we obtain their embedded forms $\boldsymbol{e}_{\boldsymbol{s}^{(i)}} = g(\boldsymbol{s}^{(i)})$ and $\boldsymbol{e}_{\boldsymbol{a}^{(j)}} = g(\boldsymbol{a}^{(j)})$.

We want to project the embedded input images to a coordinate system consisting of the embedded anchor images, and if this coordinate system is invariant to isometric transformations, we can alleviate the latent space misalignment issue. Therefore, we calculate a similarity score $r = \text{sim}(\boldsymbol{e}_{\boldsymbol{s}^{(i)}}, \boldsymbol{e}_{\boldsymbol{a}^{(j)}})$ between an embedded task state and an embedded anchor state where $sim : \mathbb{R}^d \times \mathbb{R}^d \to \mathbb{R}$. Then the relative representation (Jian et al., 2023; Moschella et al., 2022) of the input image $\boldsymbol{s}^{(i)}$ with respect to the anchor set $\mathbb{A}$ is given by:

$$\boldsymbol{r_s} = (\text{sim}(\boldsymbol{e}_{\boldsymbol{s}^{(i)}}, \boldsymbol{e}_{\boldsymbol{a}^{(1)}}), \text{sim}(\boldsymbol{e}_{\boldsymbol{s}^{(i)}}, \boldsymbol{e}_{\boldsymbol{a}^{(2)}}), \ldots, \text{sim}(\boldsymbol{e}_{\boldsymbol{s}^{(i)}}, \boldsymbol{e}_{\boldsymbol{a}^{(|A|)}})) \tag{3}$$

We choose the cosine similarity due to its invariance to reflection, rotation, and re-scaling. Additionally, we add a normalization layer before calculating cosine similarity to mitigate the translation transformation. Therefore, this relative representation is invariant to the isometric transformation, leading to better latent space alignment.

Compared with previous works (Jian et al., 2023; Moschella et al., 2022), we develop two novel techniques for the latent space alignment of visuomotor robot policies: (1) a novel anchors selection method designed for imitation learning and (2) the use of a disentanglement regularization for better latent space alignment.

**Anchors Selection.** When PeS is applied to visuomotor policies with two visual encoders, these two visual encoders encode images observed in two different cameras, and our proposed anchor selection method utilizes this correspondence between these two visual configurations. As shown in Fig. 3, after collecting a dataset $\mathbb{D}_1$ in an environment $E_1$, we perform k-means (Hartigan & Wong, 1979) on $\mathbb{D}_1$ and select the images closest to the cluster centers as the anchors in the anchor set $\mathbb{A}_1$. We then replay the trajectories of the dataset $\mathbb{D}_1$ in another environment $E_2$, which requires the robot to accomplish the same task as in $E_1$, but the cameras are different. There is no additional policy training required due to the replay. The dataset $\mathbb{D}_2$ collected by replaying these trajectories in $E_2$ has states corresponding to the states in $\mathbb{D}_1$. We then use the indices of anchors in $\mathbb{A}_1$ to select the anchor set $\mathbb{A}_2$.

**Disentanglement Regularization.** In addition to the negative log-likelihood loss ($L_{BC}$) used for the standard behavior cloning algorithm, we also apply a disentanglement regularization (Wang et al., 2022) $L_{disent}$ to further refine the latent space alignment. We first calculate the covariance of the $k^{th}$ and $l^{th}$ dimension of the batch of embedded representations with

$$\mathrm{cov}\,(z_k, z_l) = \frac{1}{N-1} \sum_{i=1}^{N} (z_{ik} - \bar{z}_k) \cdot (z_{il} - \bar{z}_l), \tag{4}$$

where $z$ is a batch of latent representations embedded by the ResNet before going through the relative representation calculation process. $z_{ik}$ and $z_{il}$ are the values of the $k^{th}$ and $l^{th}$ dimension of the $i^{th}$

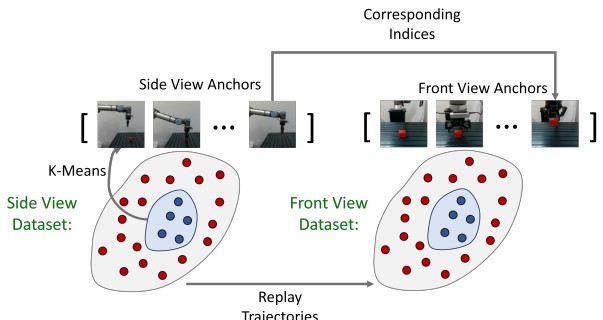

Figure 3: **Anchors Selection.** Select the anchor images in one dataset with the k-means algorithm (Hartigan & Wong, 1979). Replay the trajectories of the first dataset to collect another dataset with a different camera. We select the images with the corresponding indices of the anchors in the first dataset as the anchors in the second dataset.

embedded data point. $\bar{z}_k$ is the mean of the $k^{th}$ dimension across all $N$ data points in the batch, calculated as $\bar{z}_k = \frac{1}{N} \sum_{i=1}^{N} z_{ik}$. $\bar{z}_l$ is the mean of the $l^{th}$ dimension, similarly calculated. Then the disentanglement loss is calculated by:

$$L_{\mathrm{disent}} = \frac{1}{Z(Z-1)} \sum_{k=1}^{Z} \sum_{l=1, l\neq k}^{Z} |\mathrm{cov}\,(z_k, z_l)|, \tag{5}$$

where $Z$ is the dimensionality of the latent space. Overall, this disentanglement loss calculates the covariance of a latent representation feature $z_k$ with all other features $z_l$ ($l \neq k$), sums up these absolute values, normalizes it with $Z - 1$, and calculates the mean over all the features $z_k$ ($k = 1, 2, ..., Z$). By encouraging different features at the latent space to be independent with each other, we encourage them to capture different underlying factors hidden in the observation (e.g. object color, position etc.). The disentangled representation has been used in many applications in supervised learning (Tran et al., 2017; Kim & Mnih, 2018; Chen et al., 2018; Higgins et al., 2017; Quessard et al., 2020; Higgins et al., 2018; Zbontar et al., 2021; Ermolov et al., 2021; Bardes et al., 2021), and we empirically find it significantly improves the performance of PeS in difficult tasks.

The final PeS loss function is

$$L_{PeS} = L_{BC} + \lambda L_{disent}, \tag{6}$$

where we choose the weight $\lambda = 0.002$. For an ablation study, we also experiment without the disentanglement loss

$$L_{PeS(w/o\ disent.\ loss)} = L_{BC}. \tag{7}$$

We also experiment with replacing the disentanglement loss with $L1$ and $L2$ normalizations of the latent representations

$$L_{PeS(w.\ l1\&l2\ loss)} = L_{BC} + \lambda_1 \frac{1}{N} \sum_{i=1}^{N} \|z_i\|_1 + \lambda_2 \frac{1}{N} \sum_{i=1}^{N} \|z_i\|_2, \tag{8}$$

where the weights $\lambda_1 = 0.001$, $\lambda_2 = 0.001$, $z_i$ is the $i^{th}$ latent representation, and $N$ is the batch size. These L1 and L2 regularizations have been used in previous work (Nair et al., 2022) to avoid the state-distribution shift failure in imitation learning (Ross et al., 2011) by limiting the latent space dimension. We find it improves the zero-shot transfer performance in some cases but generally doesn't perform as well as the disentanglement loss.

### 3.3 Implementation Details

Each input image is a $(84, 84, 3)$ RGB image. The visual encoder consists of a ResNet-18 network (He et al., 2016), followed by a spatial-softmax layer (Finn et al., 2016; Mandlekar et al., 2021), and then a 256-dimensional last layer at the module interface. We use 256 anchors to match the dimension of the latent representation. The eight-dimensional proprioception consists of the end effector position (3D), end effector quaternion (4D), and the gripper open width (1D). It is embedded by a 64-dimensional linear layer and then concatenated with two latent representations of the two images. We use the RNN-based action decoder for the Can and Door Open task and the MLP-based action decoder for other tasks. We list all the parameters of the neural network in the Appendix A with our code base. The actions are output by a Gaussian Mixture Model in the last layer of the policy network where the eight-dimensional action vector is the delta value of the 8D proprioception. Before the perception stitching and zero-shot transfer, all policies are trained to 100% success rates with BC (Pomerleau, 1988). The dataset of each task contains 200 trajectories of the expert demonstrations. The pseudo code of PeS is presented in Appendix B B.

## 4 Experiments

Our experiments aim to (1) evaluate the effectiveness of PeS on enhancing zero-shot policy transfer on the stitched policies and (2) understand the mechanisms behind the performance advantage of PeS. Our experiments consist of five parts. First, we evaluate the zero-shot transfer performance of PeS for training double-camera-view visuomotor policies in five simulated manipulation tasks with seven variations of camera configurations. Next, we apply PeS to single-camera-view and triple-camera-view policies to test its generalizability to arbitrary camera number settings. We then assess the performance of PeS in four real-world manipulation tasks with three different camera configurations. In addition, we analyze the latent representations of the visual encoders both quantitatively and qualitatively to understand the effectiveness of PeS in latent space alignment. Lastly, we adopt Gradient-weighted Class Activation Mapping (Grad-CAM) (Selvaraju et al., 2017) to visualize the regions that the visuomotor policies focus on, offering an intuitive insight into the success of PeS.

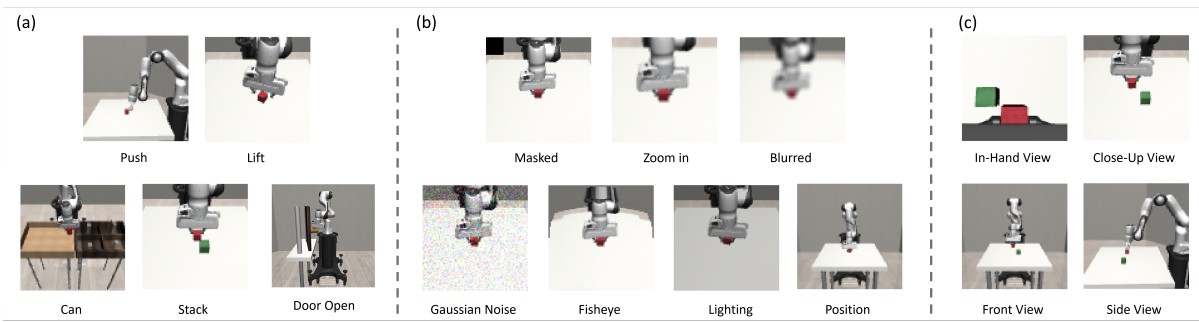

Figure 4: **Simulation Experiment Setup.** (a) Five simulation tasks from Robomimic (Mandlekar et al., 2021) benchmark. (b) Seven camera configuration variations include. (c) Four camera mounting positions.

### 4.1 Simulation Experiments with Double Camera Views

**Evaluation Tasks** We evaluate PeS in five manipulation tasks from the Robomimic (Mandlekar et al., 2021) benchmark (Fig. 4(a)): (1) Push: a robot starts from a random initial position in the air and pushes

the cube forward for a certain distance (30 cm in simulation, 20 cm in real world). (2) Lift: a robot lifts up a cube randomly placed on the table. (3) Can task: a robot picks up a can on its right and places it into the bins on its left. (4) Stack: a robot picks up the red cube on the table and stacks it on the green cube. (5) Door Open: a robot grabs the handle and pulls the door open.

**Camera Configurations** For each task, we test PeS with seven camera configurations in Fig. 4(b). (1) Masked: A black square mask is put on the upper-left corner of the image to mimic a partially occluded camera lens. Although the mask makes little difference for humans, our experiments suggest a strong negative effect on current visuomotor policies. (2) Zoom in: zoom in the camera to enlarge the object in a smaller field of view. 3) Blurred: Gaussian blur effect is added to the image. (4) Gaussian Noise: Gaussian noise is added to the image to simulate low-light condition image capture. (5) Fisheye: a fisheye effect is simulated. (6) Lighting: the image is darker or brighter than the normal images, mimicking the overexposed or underexposed situation. (7) Position: the camera is put in one out of four possible positions as in Fig. 4(c).

The first six configurations used in-hand and close-up cameras. For example, in the Push-Blurred experiment, we first train policy 1 with a normal in-hand camera and a blurry close-up camera, and policy 2 with a blurry in-hand camera and a normal close-up camera. Then we stitch the close-up view visual encoder of the policy 2 to the in-hand view visual encoder and action decoder of the policy 1 and test this stitched policy in the Push task with a normal in-hand camera and a normal close-up camera without any fine-tuning. The last configuration is designed to test zero-shot transfer ability across different camera positions. Policy 1 is trained with an in-hand camera and a front-view camera. Policy 2 is trained with a close-up view camera and a side-view camera. Then the side-view encoder of policy 2 is stitched to the in-hand view encoder and action decoder of policy 1, and tested with the in-hand and side-view cameras.

**Baselines** We adopt four baseline methods from previous works and two ablation studies: (1) RT-1 (Brohan et al., 2022) is a multi-task large model for robotic control built on the Transformer architecture. It is trained on a large dataset of various robotic tasks and can be generalized in zero-shot to new tasks. (2) Devin et al. (2017) uses a task network module to embed the observation (task state) agnostic to the robot itself, concatenates it with the proprioceptive robot state, and then inputs it into the downstream robot network module. For a fair comparison, the task module and the robot module use the same structure as our PeS method. The key difference is that Devin et al. (2017) does not adopt the relative representation and does not apply our disentanglement regularization. (3) Cannistraci et al. (2023) (linear) uses four similarity functions, $L_1$, $L_2$, $L_\infty$, $cos$ to calculate four relative representations and then calculates the element-wise linear sum of the four relative representations to be the final latent state passed over to the action decoder. The weights of the $L_1$, $L_2$, $L_\infty$, $cos$ similarities are 0.05, 0.1, 0.05, and 0.8. In contrast, our PeS method only uses cosine similarity. (4) Cannistraci et al. (2023) (nonlinear) also adopts these four similarity functions but passes them through a non-linear layer (consists of a LayerNorm, a linear layer, and a Tanh) followed by an element-wise summation. (5) The PeS (w/o disent. loss) ablation method does not have the disentanglement loss Eq. 7. (6) The PeS (w. l1 & l2 loss) replaces the disentanglement loss in PeS with L1 and L2 penalty, as shown in Eq. 8. In addition, we also report the performance of policies 1 and 2 to be stitched if they are transferred to the target environment in zero-shot. These two baselines help us understand how much this transfer preserves the performance of the policy in the initial setting, and how much extra performance increase PeS can bring.

**Results** Tab. 1 shows that PeS and its two ablation methods can all achieve satisfying performance with around 90% of success rates on Push and Lift. Compared with directly transferring policy 1 or policy 2 to the new environment, PeS results in a 25% to 45% success rate increase. RT-1 demonstrates satisfying robustness on moderate visual variations such as "Mask" and "Noise". However, RT-1 doesn't perform well when the task becomes harder and the visual configuration changes more drastically. For example, it only achieves a 14% of success rate in the Lift-Fisheye task. Cannistraci et al. 2024 (linear) has lower success rates around 70% - 80%, suggesting that introducing a linear combination of multiple similarity measurements for the relative representation hurts the model's performance. Since isometric transformation is the major transformation observed at the latent space, the single cosine similarity of PeS offers simple but effective features. Cannistraci et al. 2024 (non-linear) performs worse than the linear baseline and reaches about 20% to 50% success rates. We hypothesize that the nonlinear combination introduces learnable parameters for the non-linear summation process. However, these parameters are not optimized to align the latent spaces

|  |  | Mask | Zoom in | Blurred | Noise | Fisheye | Position | Lighting | Average |
|---|---|---|---|---|---|---|---|---|---|
| Push | Policy 1 | 94.0±2.83 | 88.7±0.94 | 88.0±1.63 | 74.7±5.73 | 69.3±0.94 | 41.3±0.94 | 38.0±1.63 | 70.6 |
|  | Policy 2 | 96.7±0.94 | 86.0±1.63 | 84.7±3.40 | 73.3±2.49 | 68.7±1.88 | 7.3±0.97 | 36.0±2.83 | 64.7 |
|  | RT-1 | 95.3±2.49 | 89.3±7.36 | 86.0±4.32 | 77.3±8.22 | 76.0±8.49 | 45.3±5.73 | 40.0±6.53 | 72.7 |
|  | Devin et al. 2017 | 60.7±10.6 | 8.7±4.99 | 16.7±3.77 | 59.3±6.80 | 29.3±7.36 | 19.3±5.73 | 36.7±3.40 | 33.0 |
|  | Cannistraci et al. 2024 (linear) | 89.3±4.11 | 94.0±2.83 | 64.7±1.89 | 74.7±6.18 | 74.0±2.83 | 78.7±2.49 | 87.3±0.94 | 80.4 |
|  | Cannistraci et al. 2024 (non-linear) | 12.7±1.89 | 18.7±4.99 | 42.8±3.27 | 23.3±0.94 | 6.0±4.32 | 5.3±2.49 | 32.7±3.40 | 20.2 |
|  | PeS (w/o disent. loss) | **100.0±0.0** | 86.0±2.83 | 80.7±9.84 | **100.0±0.0** | **100.0±0.0** | **100.0±0.0** | 86.7±0.94 | 93.3 |
|  | PeS (w. l1 & l2 loss) | 88.7±4.99 | 95.3±1.89 | 90.0±5.66 | **100.0±0.0** | 93.3±0.94 | 80.7±4.99 | 89.3±0.94 | 91.0 |
|  | PeS | **100.0±0.00** | **100.0±0.00** | **95.3±0.94** | **100.0±0.0** | 92.7±2.50 | **100.0±0.00** | **94.0±4.32** | **97.4** |
| Lift | Policy 1 | 86.0±1.63 | 24.7±3.77 | 68.0±4.90 | 88.0±1.63 | 38.7±5.25 | 0.0±0.00 | 15.3±6.18 | 45.8 |
|  | Policy 2 | 82.0±4.32 | 12.0±7.12 | 91.3±3.40 | 83.3±0.94 | 13.3±0.94 | 0.0±0.00 | 44.0±2.83 | 46.6 |
|  | RT-1 | 86.0±2.82 | 0.0±0.00 | 86.7±6.24 | 85.3±4.99 | 14.0±3.27 | 35.0±7.07 | 46.0±4.32 | 50.4 |
|  | Devin et al. 2017 | 0.0±0.00 | 5.3±2.49 | 48.0±5.89 | 9.3±4.11 | 14.7±4.99 | 36.0±1.63 | 18.7±4.99 | 18.9 |
|  | Cannistraci et al. 2024 (linear) | 72.7±3.77 | 64.0±2.83 | 86.0±4.32 | 88.7±1.88 | 57.3±2.49 | 56.0±5.89 |  | 70.5 |
|  | Cannistraci et al. 2024 (non-linear) | 89.3±2.49 | 36.0±3.27 | 52.7±3.40 | 93.3±2.49 | 16.7±2.49 | 21.3±0.94 | 37.3±2.49 | 49.5 |
|  | PeS (w/o disent. loss) | 83.3±6.60 | 80.7±5.73 | **93.3±0.94** | 91.3±5.73 | 79.3±2.49 | **93.3±2.49** | 76.7±3.40 | 85.4 |
|  | PeS (w. l1 & l2 loss) | **97.3±2.49** | 85.3±0.94 | 90.7±0.94 | 86.0±4.32 | 88.0±1.63 | 84.7±3.77 | 64.7±3.40 | 85.2 |
|  | PeS | 92.7±2.50 | **94.7±1.89** | 89.3±4.11 | **96.0±1.63** | **88.7±0.94** | 93.0±0.03 | **84.7±6.60** | **91.3** |

Table 1: **Zero-Shot Transfer Success Rates in basic Simulation tasks.** We test all methods across six different visual configurations and also calculate the average performance. In the two basic tasks, Push and Lift, PeS and its two ablation methods can all get about a 90% success rate on average. The Cannistraci et al. 2024 (linear) baseline has 70% - 80% of success rate, while the Cannistraci et al. 2024 (non-linear) and Devin et al. 2017 baselines perform poorly with about 20% - 50% of success rate.

|  |  | Mask | Zoom in | Blurred | Noise | Fisheye | Position | Lighting | Average |
|---|---|---|---|---|---|---|---|---|---|
| Can | Policy 1 | 76.7±3.40 | 12.7±0.94 | 80.0±2.83 | 83.3±2.49 | 16.0±1.63 | 0.0±0.00 | 0.0±0.00 | 38.4 |
|  | Policy 2 | 80.0±0.00 | 4.0±1.63 | 78.0±0.00 | 86.7±0.94 | 13.3±4.11 | 0.0±0.00 | 2.0±0.00 | 37.7 |
|  | RT-1 | **88.3±6.24** | 0.0±0.00 | **86.7±8.50** | **88.3±2.36** | 0.0±0.00 | 8.3±6.24 | 4.0±0.00 | 39.4 |
|  | Devin et al. 2017 | 19.3±5.25 | 24.7±1.89 | 2.7±1.89 | 6.0±4.32 | 29.3±3.40 | 1.3±1.89 | 3.3±1.89 | 12.4 |
|  | Cannistraci et al. 2024 (linear) | 33.3±0.94 | 48.0±1.63 | 48.7±2.49 | 65.3±0.94 | 26.7±3.77 | 34.7±3.77 | 42.7±0.94 | 42.8 |
|  | Cannistraci et al. 2024 (non-linear) | 72.7±0.94 | 24.7±2.49 | 37.3±4.99 | 42.7±3.40 | 8.7±1.89 | 39.3±1.89 | 38.7±1.89 | 37.7 |
|  | PeS (w/o disent. loss) | 44.7±8.06 | **89.3±4.11** | 34.7±4.11 | 30.7±6.80 | **92.7±2.50** | 44.7±3.40 | 42.7±0.94 | 54.2 |
|  | PeS (w. l1 & l2 loss) | 47.3±0.94 | 58.7±1.88 | 54.0±8.64 | 36.0±7.12 | 58.7±1.88 | 64.7±6.60 | 38.0±4.32 | 51.1 |
|  | PeS | 83.3±5.24 | 89.3±2.49 | 74.0±2.83 | 78.7±4.11 | 56.0±2.83 | **78.7±2.49** | **73.3±6.80** | **76.2** |
| Stack | Policy 1 | 66.0±2.83 | 16.0±5.89 | 1.3±0.94 | 65.3±0.94 | 26.7±2.49 | 0.0±0.00 | 15.3±6.18 | 27.2 |
|  | Policy 2 | 83.3±2.49 | 0.0±0.00 | 4.0±1.63 | 64.0±0.00 | 2.0±1.63 | 2.0±1.63 | 24.0±2.83 | 25.6 |
|  | RT-1 | 85.0±4.08 | 0.0±0.00 | 0.0±0.00 | 76.7±4.71 | 13.3±5.73 | 0.0±0.00 | 31.3±0.94 | 29.5 |
|  | Devin et al. 2017 | 0.7±0.94 | 8.0±1.63 | 0.7±0.94 | 24.0±2.83 | 0.0±0.00 | 14.0±3.27 | 4.7±2.49 | 7.4 |
|  | Cannistraci et al. 2024 (linear) | 47.3±0.94 | 62.0±4.32 | 32.7±3.77 | 30.7±0.94 | 54.0±8.64 | 14.7±6.18 | 0.7±0.94 | 34.6 |
|  | Cannistraci et al. 2024 (non-linear) | 10.0±1.63 | 12.0±0.00 | 0.0±0.00 | 3.3±0.94 | 0.0±0.00 | 0.7±0.94 | 8.7±3.77 | 5.0 |
|  | PeS (w/o disent. loss) | 34.0±11.43 | 10.7±4.11 | 62.0±10.71 | 34.0±7.12 | 22.7±3.77 | 26.0±4.32 | 36.7±8.06 | 32.3 |
|  | PeS (w. l1 & l2 loss) | 92.7±0.94 | **98.0±0.00** | 62.7±6.60 | 24.0±4.90 | 59.3±7.36 | 58.7±1.88 | 48.0±4.32 | 63.3 |
|  | PeS | **94.7±0.94** | 96.7±0.94 | **90.0±1.63** | **96.7±1.89** | **97.3±2.49** | **80.0±4.90** | **82.0±1.63** | **91.1** |
| Door Open | Policy 1 | 95.3±1.89 | 0.0±0.00 | 19.3±4.71 | 66.0±4.32 | 4.0±1.63 | 0.0±0.00 | 0.7±0.94 | 26.5 |
|  | Policy 2 | 89.3±0.94 | 5.3±1.89 | 21.3±5.25 | 52.7±0.94 | 4.0±1.63 | 0.0±0.00 | 14.7±3.40 | 26.8 |
|  | RT-1 | **96.0±1.63** | 0.0±0.00 | 26.7±3.40 | **94.7±3.40** | 0.7±0.94 | 0.0±0.00 | 0.0±0.00 | 31.2 |
|  | Devin et al. 2017 | 9.3±4.11 | 5.3±0.94 | 0.0±0.00 | 4.0±1.63 | 0.7±0.94 | 0.0±0.00 | 6.0±1.63 | 3.6 |
|  | Cannistraci et al. 2024 (linear) | 0.0±0.00 | 1.3±0.94 | 10.7±2.49 | 10.7±4.99 | 2.0±1.63 | 47.3±9.29 | 12.0±7.12 | 12.0 |
|  | Cannistraci et al. 2024 (non-linear) | 26.0±2.83 | 31.3±4.99 | 49.3±8.22 | 48.0±5.89 | 62.7±3.40 | 44.7±3.40 | 33.3±4.99 | 42.2 |
|  | PeS (w/o disent. loss) | 24.7±7.71 | 44.0±2.83 | 34.7±3.77 | 0.7±0.94 | 36.7±0.94 | 23.3±3.40 | 24.0±2.83 | 26.9 |
|  | PeS (w. l1 & l2 loss) | 4.0±1.63 | **78.0±5.66** | 3.3±0.94 | 2.0±1.63 | 42.7±4.99 | 6.0±3.26 | 14.7±4.99 | 21.5 |
|  | PeS | 58.7±4.11 | 68.7±0.94 | **70.7±0.94** | 52.7±3.40 | **64.7±4.99** | **48.7±3.40** | **56.7±5.73** | **60.1** |

Table 2: **Zero-Shot Transfer Success Rates in difficult Simulation tasks.** The three difficult tasks include Can, Stack, and Door Open. We test all methods across six different visual configurations and also calculate the average performance. The three baselines achieve about 0% - 40% of success rates in these tasks. In the Can and Stack tasks, PeS achieves 75% - 95% of success rates, while the ablation methods achieves about 30% - 65% of success rates. In the Door task, PeS achieves 60.7% of success rates, while the two ablation methods only have 20% - 30% of success rates.

but are optimized for the accuracy of behavior cloning, leading to unaligned latent space and hurting the performance. Devin et al. 2017 baseline achieves very low success rates between 18% to 33%, indicating that the unaligned latent representations cause the failure of the stitched policy for zero-shot transfer.

Tab. 2 reports the performance in more difficult tasks: Can, Stack, and Door Open. PeS achieves 60% to 93% of success rates, which has about 30% to 60% of advantage to the base policy 1 and 2. The success rates of all four baselines are lower than 43%. This result suggests that previous methods cannot solve the zero-shot transfer problem for difficult tasks with satisfactory performance, which usually involve complicated visual background (e.g., Can), long-horizon multi-stage motions (e.g., Stack), and articulated objects (e.g., Door Open). Among these baselines, we find that RT-1 sometimes can achieve very good performance when the visual variation is moderate. For example, in the Can-Mask task and the Door Open-Noise task,

RT-1 achieves better performance than all other methods including PeS. This is somewhat not surprising since RT-1 was trained on a much larger dataset with a much larger model that could cover some of these variations in training. However, for the harder tasks with more drastic visual configuration changes, such as "Door-Fisheye" and "Stack-Camera Position", we found that the RT-1 method could not perform well. This additional result suggests that the trade-off between modular and end-to-end architectures is that an end-to-end network is simpler in the end-to-end training procedure but usually cannot generalize to drastic changes in visual configuration that have not been encountered during training. In contrast, although the modular policies usually involve more careful structure designs in the training procedure, our results show that they performed better in drastic visual changes.

We also notice that the two ablation methods have significant performance drops (20% to 60%) in these difficult tasks, showing that the disentanglement regularization in PeS largely improves the zero-shot transfer performance. Although using L1 and L2 regularization also leads to good success rates in some cases, the disentanglement regularization has more consistent performance and higher success rates on average.

## 4.2 Simulation Experiments with Single Camera View and Triple Camera Views

While most visuomotor policies adopt two cameras in their applications, we also conduct experiments for policies with only one camera and policies with three cameras to verify the generalizability of PeS to arbitrary camera number settings.

**Single Camera View** We train two policies with two different cameras separately. Each policy still has two visual encoders, but one visual encoder takes in a black image and the other encoder takes in the images from the single camera. For both encoders of the policy, they use the same anchor images set collected by the only camera, because we assume that each policy is trained separately and only has access to the dataset of one camera.

In the perception stitching process, we stitch the visual encoder of policy 2 to the visual encoder and action decoder of policy 1, as shown in Figure 5, and assemble a stitched policy for double camera inputs. We suggest that this is the most reasonable setting for the perception stitching of single-camera policies. In contrast, if we remove the black image encoder and stitch the only vision encoder of the policy 1 to the action decoder of the policy 2, applying this reassembled policy in either environment 1 or 2 won't lead to better performance compared with the original policy trained to a maximum success rate in that environment.

As shown in table 3, PeS outperforms the other methods in the three difficult tasks and achieves close-to-optimal performance in the two basic tasks. It has the highest average success rate over the five tasks.

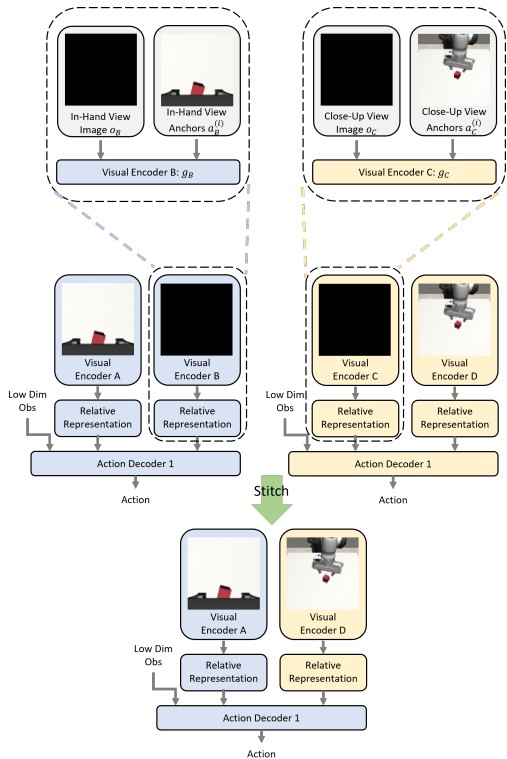

Figure 5: **Perception Stitching with Single Camera.** The original two policies are trained with only one camera. The other visual encoder takes in a black image. The corresponding anchors are selected with our proposed method 3. The visual encoder of the black image uses the same anchor images as the other encoder of this policy.

Ideally, the decoder should learn to completely ignore the latent representations from the encoder with empty input if we train the policy with a large amount of expert data and many epochs. In practice, however, the policy is trained with only 200 expert demonstrations of data and limited epochs. The policy will learn to assign more importance to the encoder with RGB image input and less importance to the encoder with empty input, but it will not fully ignore that encoder with empty input. Therefore, stitching another encoder to replace the encoder with empty input will have less influence on the policy performance compared with the double camera experiments, but the disturbance caused by this new encoder will still exist due to the latent

space misalignment, although in some easier experiments, the disturbance is marginal. As supported by our single-camera experiment results, the performance decreases of the baseline methods after the zero-shot transfer are smaller, compared with that in the double-camera experiments, but the performance decreases are not zero.

| | Push | Lift | Can | Stack | Door Open | Average |
|---|---|---|---|---|---|---|
| Devin et al. 2017 | 73.3±11.47 | 72.7±0.94 | 66.7±6.18 | 14.0±3.27 | 22.7±2.50 | 49.8 |
| Cannistraci et al. 2024 (linear) | 89.3±4.11 | 86.0±1.63 | 72.8±3.27 | 14.7±6.18 | 39.3±3.40 | 60.4 |
| Cannistraci et al. 2024 (non-linear) | 60.7±1.88 | 80.7±3.40 | 24.0±2.83 | 0.7±0.94 | 41.3±4.99 | 41.5 |
| PeS (-w/o disent. loss) | 88.7±4.11 | **94.0±1.63** | 91.0±3.40 | 26.0±4.32 | 32.7±3.77 | 66.5 |
| PeS (w. l1 & l2 loss) | **94.7±2.49** | 90.0±2.83 | 86.7±1.89 | 58.7±1.88 | 40.7±0.94 | 74.2 |
| PeS | 91.3±2.49 | 92.6±0.94 | **94.6±0.94** | **80.0±4.90** | **72.7±3.77** | **86.2** |

Table 3: **Zero-Shot Transfer Success Rates of single camera policies.** PeS achieves optimal performance in the three difficult tasks (Can, Stack, Door Open), and close-to optimal performance in the two basic tasks (Push, Lift). Its average success rate outperforms all the baselines and the ablation methods.

| | Mask | Zoom in | Blurred | Noise | Fisheye | Camera Position | Average |
|---|---|---|---|---|---|---|---|
| Devin et al. 2017 | 4.0±1.63 | 5.3±2.49 | 1.3±0.94 | 13.3±1.89 | 1.3±1.89 | 8.0±1.63 | 5.5 |
| Cannistraci et al. 2024 (linear) | 46.7±3.40 | 20.7±1.89 | 13.3±1.89 | 56.0±2.83 | 16.0±4.32 | 30.0±4.32 | 30.5 |
| Cannistraci et al. 2024 (non-linear) | 30.7±0.94 | 22.7±0.94 | 19.3±4.11 | 42.7±1.89 | 12.7±0.94 | 4.7±2.49 | 22.1 |
| PeS (-w/o disent. loss) | 21.3±2.49 | 20.7±1.89 | 72.7±3.77 | 56.7±4.11 | 37.3±4.99 | 9.3±1.89 | 36.3 |
| PeS (w. l1 & l2 loss) | 46.7±3.40 | 56.7±4.11 | 84.0±2.83 | 82.7±2.49 | 75.3±2.49 | 61.3±3.40 | 67.8 |
| PeS | **97.3±2.49** | **92.7±0.94** | **93.3±0.94** | **90.0±1.63** | **94.6±0.94** | **86.7±2.49** | **92.4** |

Table 4: **Zero-Shot Transfer Success Rates of triple camera policies.** All the experiments are carried out with the Stack task. PeS achieves optimal performance for all the six different visual configurations. Its average success rate has an around 25% advantage to the second best method.

**Triple Camera Views** We pick the Stack task and train policies with three visual encoders that embed three different visual configurations. For the first five visual configurations that add different effects to the observations, policies are all trained with the in-hand view, close-up view, and side view. Policy 1 has the effect (e.g. Gaussian noise) only in vision 3, and policy 2 has the same effect in vision 1 and vision 2. Then we stitch the encoder 3 of policy 2 to the rest parts of policy 1 and test the stitched policy with three normal cameras. For the last experiment that involves different camera positions, policy 1 is trained with the in-hand view, close-up view, and front view, and policy 2 is trained with close-up view, front view, and side view. Then we stitch the side view encoder of policy 2 to the in-hand view, close-up view encoders, and decoder of policy 1. The stitched policy is tested with in-hand view, close-up view, and side view. This triple camera perception stitching process is presented in Figure 6.

Table 4 reports the zero-shot transfer success rates for the Stack task with six different visual configurations. The PeS method consistently outperforms the other baseline and ablation methods for all six visual configurations. On average, it reaches a 92.4% success rate on the stacking task, which is about 25% higher than the second-best method.

### 4.3 Real World Experiment

**Camera Configurations** We conducted Reach, Push, Lift, and Stack in our real-world experiments (Fig. 7). In Reach, the robot starts from some random position in the air and reaches the red cube. In this experiment, the policy 1 is trained with a front-view camera with a broken lens (Fig. 7) and a normal in-hand camera. The policy 2 is trained with a normal front-view camera and an in-hand camera with a

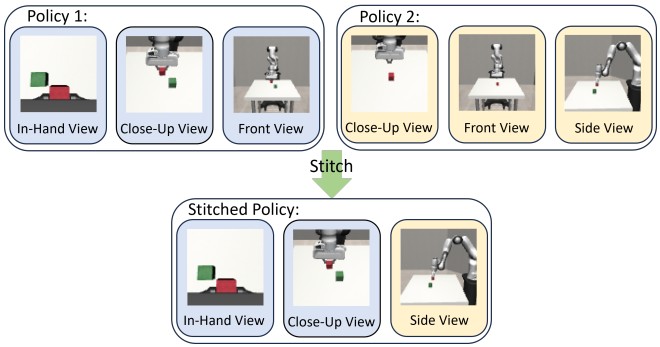

Figure 6: **Perception Stitching with Three Cameras.** The original two policies are trained with three cameras. In the Perception Stitching process, the visual encoder 3 of the policy 2 is stitched to the visual encoder 1 and 2 and the action decoder of the policy 1.

broken lens. Then we stitch the front view encoder of the policy 2 to the in-hand view encoder and action decoder of the policy 1 and test the stitched policy with two normal cameras. In Push, the robot starts from some random position to reach the cube and then pushes the cube across the green line on the table. The training and testing process of Push is similar to that of Reach, and the only difference is replacing the camera with a broken lens with a camera with a masked lens, as shown in Fig. 7. In Lift, the red cube is placed at a random position on the table with a random orientation. The robot needs to grab the cube and lift it up. In this experiment, the policy 1 is trained with in-hand and side-view cameras. The policy 2 is trained with side-view and front-view cameras. Then we stitch the front view encoder of the policy 2 to the in-hand view encoder and action decoder of the policy 1, and test the stitched policy with an in-hand camera and a front-view camera, as shown in the left half part of the Fig. 2. In Stack, the red cube is placed at a random position on the left side of the table, and the robot should lift it up and stack it on the blue cube on the right. Similar to Lift, it also involves the stitching between cameras at different positions (left, front, and right). Stack is the most difficult one in all the real-world experiments. It requires the most sophisticated manipulation skills and long-horizon multi-stage motions.

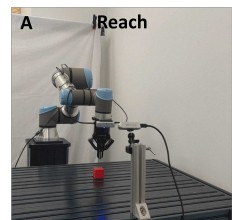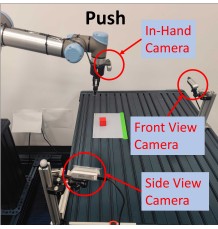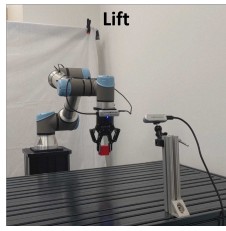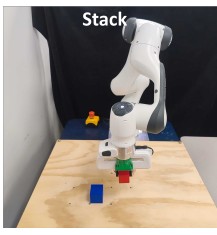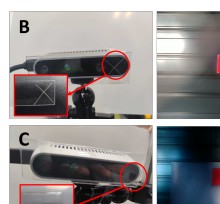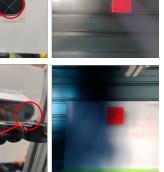

Figure 7: (A) Real-World Tasks: reaching, pushing, lifting, and stacking a cube. Three cameras are mounted at the front view, side view, and in-hand positions separately. (B) Broken lens. (C) Masked lens.

**Results** As shown in Tab. 5, PeS achieves 100% of success rate in Reach, 85% in Push, 80% in Lift, and 45% in Stack, while Devin et al. 2017 baseline gets 0% of success rates in all the experiments. In the experiments, we observe that PeS policies show more accu-

|  | Reach
broken lens | Push
masked lens | Lift
different positions | Stack
different positions |
|---|---|---|---|---|
| PeS | **100.0** | **85.0** | **80.0** | **45.0** |
| Devin et al. 2017 | 0.0 | 0.0 | 0.0 | 0.0 |

Table 5: **Zero-Shot Transfer Success Rates in Real World.** We report the success rates of three manipulation tasks in real world. Each success rate is calculated with 20 games. PeS achieves a 100% success rate in the easiest Reach task, 85% in the Push task, 80% in the Lift task, and 45% in the hardest Stack task. In comparison, the baseline method doesn't have any success case in all these tasks.

rate motions compared with the baseline policies. In comparison, Devin et al. 2017 baseline policies cannot output actions accurately enough to accomplish the tasks, although they have some attempts to complete the tasks in some cases. Take the Lift task as an example, the robot trained with the baseline method reaches some positions close to the cube but always fails to grasp it. In the most difficult Stack task, PeS robot can still achieve many success cases, while the baseline robot cannot output meaningful motions. Please refer to the supplementary material for the experiment videos.

In summary, we find that the performance advantage of PeS is more pronounced in the real world than in the simulation. We assume that this is because there are more noises and disturbances in the real world. To the best of our knowledge, PeS is the first method that enables vision-based zero-shot transfer in the real world via reassembling neural network components.

## 4.4 Analysis: Latent Space at Module Interface

To understand the mechanism behind the high success rate of PeS, we perform visualization and quantitative analysis of the latent representations of the visual encoders.

We choose the Push-Camera Position experiment in the simulation and visualize the corresponding latent representations to have an intuitive understanding of the mechanism of PeS. We first reduced the 256D representations to 3D with PCA(Hotelling, 1933) for visualization. Since the side view encoder of the policy 2 is stitched to the policy 1 at the position of its original front view encoder, we compare the latent representations of these two encoders. As shown in Fig. 8(a), the latent representations trained with PeS have

similar shapes to each other. In comparison, the latent representations with the Devin et al. 2017 baseline are not similar but have an approximately isometric transformation (rotation in this case) relationship with each other. The latent representations visualization of other tasks can be found in Appendix F.

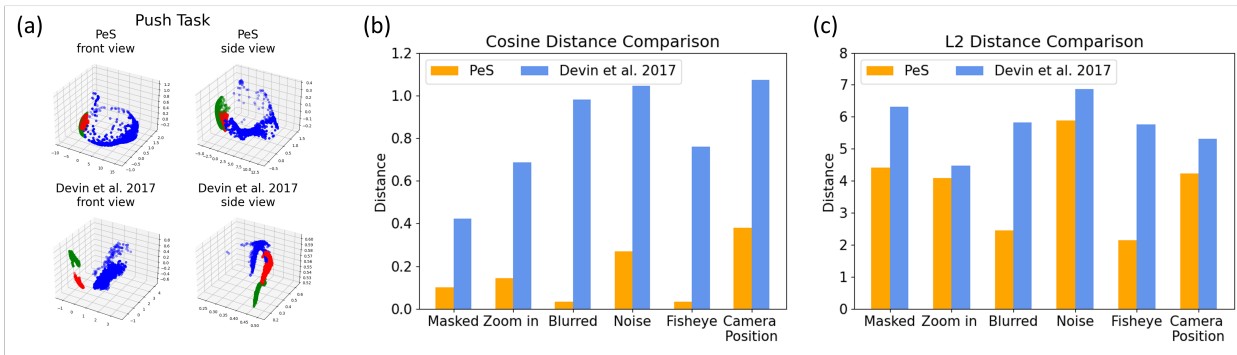

Figure 8: **Latent Space Analysis.** (a) We visualize the latent representations of the front-view encoder of policy 1 and the side-view encoder of policy 2 of the Push-Camera Position experiment. The 256D representations are reduced to 3D with PCA (Hotelling, 1933). The red dots represent samples where the robot's end effector is at higher positions, blue dots indicate medium heights and green dots correspond to lower positions near the cube. We compare our (PeS) method with the baseline (Devin et al. 2017) method. (b) & (c) We compare the distances of the latent representations in all the experiments in the Push task. One representation is from the second view encoder of policy 1 and the other is from the second view encoder of policy 2.)

For further quantitative analysis, we select the Push task in the simulation and calculate the pairwise distances of the latent representations in all Push experiments with different visual configurations. Fig. 8 (b) shows that PeS significantly reduces the cosine distances of the latent representations in all these experiments. Fig. 8 (c) shows that the L2 distances with PeS are generally smaller than that with the Devin et al. 2017 baseline, but the differences are not distinguishable in some cases (e.g. Push-Zoom in). More mathematical details of calculating the cosine and L2 distances can be found in Appendix C.

### 4.5 Analysis: Highlight Attention Regions with Grad-CAM

We visualize the attention map from the policy modules to further explain why certain policies work well while others fail. To this end, we modify the Gradient-weighted Class Activation Mapping (Grad-CAM) (Selvaraju et al., 2017) approach to highlight the regions that the policies pay attention to. Grad-CAM is widely applied to neural classification models with convolutional layers. To adapt the Grad-CAM from image classification to robot learning, we replace the before-softmax score $y^c$ for class $c$ of the image classification net-

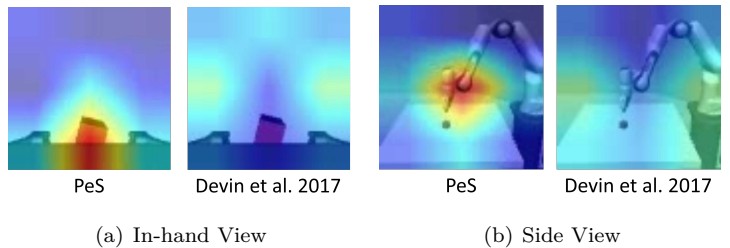

(a) In-hand View  (b) Side View

Figure 9: **Attention Heatmap.** (a) In the in-hand view, our policy pays attention to the cube on the table, while the baseline policy pays meaningless attention to the upper region of the image. (b) In the front view image, our policy pays attention to the robot end effector, while the baseline method has slightly more attention to the two sides of the image.

works with the log-likelihood $l(a)$ of the robot action $a$ in the training dataset. We denote the $k^{th}$ feature map activation output from the last convolutional layer as $A^k$. Then the backpropagated gradient of $l(a)$ with respect to $A^k$ is computed as $\frac{\partial l(a)}{\partial A^k}$. We do global average pooling of these gradients over the width (indexed by $i$) and height (indexed by $j$) dimensions of the feature map to get the neuron importance weight :

$$\alpha_k^a = \overbrace{\frac{1}{Z}\sum_i \sum_j}^{\text{global average pooling}} \underbrace{\frac{\partial l(a)}{\partial A_{ij}^k}}_{\text{gradients via backprop}} . \tag{9}$$

This weight $\alpha_k^a$ captures the "importance" of feature map $k$ for robot action $a$. Then, the attention map $L_{\text{Grad-CAM}}^a$ is calculated as the weighted combination of forward activation maps followed by a ReLU (Selvaraju et al., 2017):

$$L_{\text{Grad-CAM}}^a = \text{ReLU} \underbrace{\left( \sum_k \alpha_k^a A^k \right)}_{\text{linear combination}} . \tag{10}$$

We apply ReLU because we are only interested in the features that have a positive influence on the actions. The intensity of these pixels should be increased in order to increase the log-likelihood $l(a)$ (Selvaraju et al., 2017). This $L_{\text{Grad-CAM}}^a$ is a heatmap of the same size as the convolutional feature maps $A^k$. We upsample it to the input image size with bilinear interpolation to get the final attention heatmap of the input image (Selvaraju et al., 2017). A larger value on this heatmap means this pixel contributes to a larger gradient of the log-likelihood of the robot action.

We choose the Lift-Camera Position experiment and visualize the attention heatmap of the stitched policies with PeS and the baseline separately. As shown in Fig. 9(a), from the in-hand camera, our policy is focusing on the cube between the two gripper fingers, which is intuitively what humans pay attention to in a Lift task. In comparison, the Devin et al. 2017 baseline policy has more attention to the upper middle part of the image, which is the desk surface, and is not informative for this task. In Fig. 9(b), our policy is paying attention to the robot end effector from the side-view camera, while the baseline policy has more attention to the two sides of the image, while these regions have no crucial object. To sum up, the attention maps from the Grad-CAM suggest that the stitched policy with PeS performs better than the baseline because it can pay attention to the crucial regions for accomplishing the task, while the baseline policy cannot. This result provides an intuitive explanation of the good performance of PeS. Videos of the attention heatmaps during the manipulation process can be found in the supplementary material.

## 5 Discussion

**Conclusion** We present Perception Stitching (PeS), a method for zero-shot visuomotor policies transfer via latent spaces alignment. PeS aligns the latent spaces of different visual encoders by enforcing the relative representations invariant to isometric transformations and, therefore, allows the trained visual encoders to be reused in a plug-and-go manner. Our evaluation covers 35 simulation experiments and 4 real-world experiments with a variety of manipulation tasks and camera configurations. The results demonstrate significant performance improvement with PeS for zero-shot transfer, and its advantage is especially pronounced in real-world tasks. Moreover, we conduct quantitative and qualitative analyses to understand the mechanism of the superior performance of PeS. We hope that this work can inspire further exploration of the compositionality and modularity in robot learning.

**Limitations and Future Work** There is no additional training required during transfer, but to obtain the anchor states our framework still requires the robot to replay the trajectories in the previous dataset to collect a new dataset and then select anchors with the corresponding indices. Although this is feasible, as we have demonstrated in our real-world experiments, the trajectories replaying process in the real world usually takes longer time than collecting a new dataset by random sampling. Future work can develop an alternative algorithm for more efficient anchor selection. We also acknowledge that PeS cannot yet achieve perfect performance on some tasks that require very high precision or have very long horizons. One major cause of the potential failure is that the PeS can only enforce an approximate invariance of the latent representations but not a strict invariance, which leads to the loss of some accuracy of the stitched policy. It is an interesting topic for future work to further refine the latent space alignment.

In addition, the policy trained with the Behavior Cloning algorithm usually cannot recover from the error accumulation very well in long-horizon tasks. Although our paper focuses on imitation learning, one exciting future direction is to explore our techniques in a reinforcement learning setup (Ricciardi et al., 2024) or explore more advanced imitation learning that handles error accumulation, since these approaches have shown some capabilities of recovery from failures in recent works. Another promising direction is to explore the combination of PeS with other robot transfer learning techniques that focus beyond vision encoder transfer, such as embodiment transfer and scale to large robot learning datasets.

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

# A   Network Structure

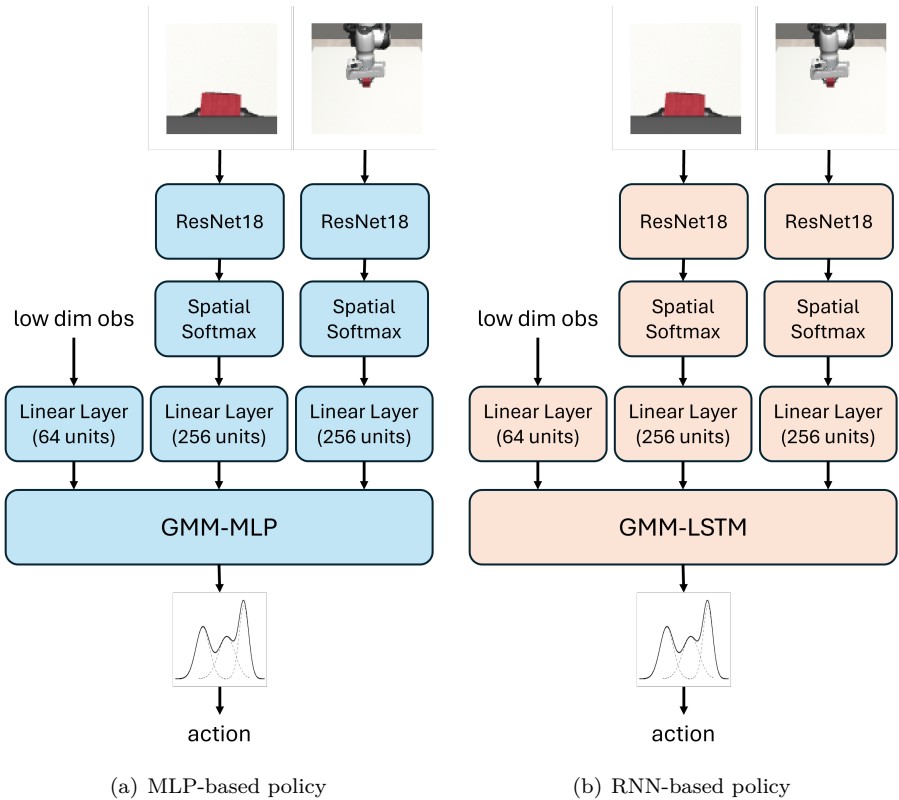

(a) MLP-based policy                    (b) RNN-based policy

Figure 10: **Policy Network Structure.** The detailed architectures of the visuomotor policy networks. We adopt the RNN-based network for the Pick-And-Place-Can task, and the MLP-based network for other tasks.

| Hyperparameter | Default |
| --- | --- |
| Learning Rate | $1 \times 10^{-4}$ |
| Action Decoder MLP Dims | [1024, 1024] |
| GMM Num Modes | 5 |
| Image Encoder | ResNet-18 |
| SpatialSoftmax (num-KP) | 64 |
| Image Embedding Layer | 256 units |
| Low Dim Obs Embedding Layer | 64 units |

Table 6: **MLP-based policy Hyperparameters.**

For the baseline methods, we demonstrate the detailed structures of the MLP-base policy network in Fig. 10(a), and the RNN-based policy network in Fig. 10(b). The hyperparameters of the MLP-base network and the RNN-based network are listed in TABLE 6 and TABLE 7 separately.

The policy network of the Perception Stitching method shares all the network structures and hyperparameters as that of the baseline method. The only difference is that it calculates the relative representations of the images after the 256-unit linear layer, as shown in Fig. 2 and equation 3.

# B   Perception Stitching Algorithm

The algorithm of the Perception Stitching (PeS) is shown in Algorithm 1. The first step of this algorithm is to collect an expert dataset $\mathbb{D}_1$ in the environment $E_1$ by teleoperating the robot by a proficient human expert.

---

**Algorithm 1** Zero-shot Transfer with Perception Stitching

---

**Collect Dataset 1 with random sampling:**

Task $T$ in the environment $E_1$ with two visual configurations $o_1^{E_1}$ and $o_2^{E_1}$. Initialize an empty dataset $\mathbb{D}_1 \leftarrow \emptyset$.

**for** each game $i$ of the task **do**
    Random sample the initial state of the task.
    Execute the Expert policy to collect the expert trajectory $\tau_i^1$ of this game.
    Push $\tau_i^1$ into the dataset $\mathbb{D}_1$.
**end for**

**Collect Dataset 2 with trajectories replay:**

Task $T$ in the environment $E_2$ with two visual configurations $o_1^{E_2}$ and $o_2^{E_2}$. Initialize an empty dataset $\mathbb{D}_2 \leftarrow \emptyset$.

**for** each game $i$ of the task **do**
    Replay the trajectory $\tau_i^1$ in $E_2$ to collect the trajectory $\tau_i^2$ of this game.
    Push $\tau_i^2$ into the dataset $\mathbb{D}_2$.
**end for**

**Collect Anchor States:**

Task $T$ in the environment $E_3$ with two visual configurations $o_1^{E_1}$ and $o_2^{E_2}$.

K-means center set $\mathbb{C}_1 \leftarrow k - means(\mathbb{D}_1)$.

Select anchor set $\mathbb{A}_1 = \{a_1^{(j)}\}$ from $\mathbb{D}_1$ which are images closest to the K-means center set $\mathbb{C}_1$.

Select anchor set $\mathbb{A}_2 = \{a_2^{(j)}\}$ from $\mathbb{D}_2$ which have the same indices as $\mathbb{A}_1$ in $\mathbb{D}_1$.

Anchor set $\mathbb{A}_3 = \{a_3^{(j)}\}$ consists of images of $o_1^{E_1}$ from $\mathbb{A}_1$ and images of $o_2^{E_2}$ from $\mathbb{A}_2$.

**Train Modular Policies in Source Environments:**

Environment $E_1$ has two visual configurations $o_1^{E_1}$, $o_2^{E_1}$ and low dimensional observation $o_l^{E_1}$.

Environment $E_2$ has two visual configurations $o_1^{E_2}$, $o_2^{E_2}$ and low dimensional observation $o_l^{E_2}$.

Initialize policy $f^{E_1}(g_1^{E_1}(o_1^{E_1}), g_2^{E_1}(o_2^{E_1}), o_l^{E_1}, \mathbb{A}_1)$ in $E_1$.

Initialize policy $f^{E_2}(g_1^{E_2}(o_1^{E_2}), g_2^{E_2}(o_2^{E_2}), o_l^{E_2}, \mathbb{A}_2)$ in $E_2$.

Optimize $f^{E_1}(g_1^{E_1}(o_1^{E_1}), g_2^{E_1}(o_2^{E_1}), o_l^{E_1}, \mathbb{A}_1)$ with dataset $\mathbb{D}_1$ with the Behavior Cloning algorithm Schaal (1996).

Optimize $f^{E_2}(g_1^{E_2}(o_1^{E_2}), g_2^{E_2}(o_2^{E_2}), o_l^{E_2}, \mathbb{A}_2)$ with dataset $\mathbb{D}_2$ with the Behavior Cloning algorithm Schaal (1996).

**Perception Stitching:**

Environment $E_3$ has visual configurations $o_1^{E_1}$ and $o_2^{E_2}$.

Initialize the visual encoder 1 with parameters from $g_1^{E_1}$.

Initialize the visual encoder 2 with parameters from $g_2^{E_2}$.

Initialize the action decoder with parameters from $f^{E_1}$.

Construct the stitched policy $f^{E_1}(g_1^{E_1}(o_1^{E_1}), g_2^{E_2}(o_2^{E_2}), o_l^{E_1}, \mathbb{A}_3)$.

**Test the Stitched Policy in the Target Environment:**

Rollout the stitched policy $f^{E_1}(g_1^{E_1}(o_1^{E_1}), g_2^{E_2}(o_2^{E_2}), o_l^{E_1}l, \mathbb{A}_3)$ in $E_3$.

Calculate the success rate of task $T$ with the policy $f^{E_1}(g_1^{E_1}(o_1^{E_1}), g_2^{E_2}(o_2^{E_2}), o_l^{E_1}, \mathbb{A}_3)$ in $E_3$.

---

| Hyperparameter | Default |
|---|---|
| Learning Rate | $1 \times 10^{-4}$ |
| Action Decoder MLP Dims | [ ] |
| RNN Hidden Dim | 1000 |
| RNN Seq Len | 10 |
| GMM Num Modes | 5 |
| Image Encoder | ResNet-18 |
| SpatialSoftmax (num-KP) | 64 |
| Image Embedding Layer | 256 units |
| Low Dim Obs Embedding Layer | 64 units |

Table 7: **RNN-based policy Hyperparameters.**

Then we replay the expert trajectories in $\mathbb{D}_1$ to collect another expert dataset $\mathbb{D}_2$ in the environment $E_2$. We collect the anchor set $\mathbb{A}_1$ in the dataset $\mathbb{D}_1$ via the K-means algorithm Hartigan & Wong (1979). Then we collect the anchor set $\mathbb{A}_2$ which has the same indices in $\mathbb{D}_2$ as $\mathbb{A}_1$ in $\mathbb{D}_1$. For the environment $E_3$ with two visual configurations $o_1^{E_1}$ and $o_2^{E_2}$, we assemble an anchor set $\mathbb{A}_3 = \{a_3^{(j)}\}$ consists of images of $o_1^{E_1}$ from $\mathbb{A}_1$ and images of $o_2^{E_2}$ from $\mathbb{A}_2$. In the next step, we train the two polices $f^{E_1}(g_1^{E_1}(o_1^{E_1}), g_2^{E_1}(o_2^{E_1}), o_l^{E_1}, \mathbb{A}_1)$ and $f^{E_2}(g_1^{E_2}(o_1^{E_2}), g_2^{E_2}(o_2^{E_2}), o_l^{E_2}, \mathbb{A}_2)$ in $E_1$ and $E_2$ with the Behavior Cloning algorithm Schaal (1996) separately. Then we initialize a stitched policy $f^{E_1}(g_1^{E_1}(o_1^{E_1}), g_2^{E_2}(o_2^{E_2}), o_l^{E_1}, \mathbb{A}_3)$. This stitched policy is tested in the task $T$ in $E_3$. Its performance is measured by its success rate.

## C    Quantitative Analysis of the Module Interface

The cosine and L2 pairwise distance shown in Fig. 8 measures the similarity between two latent representations. For a group of states $\mathbb{S}_{E,T} = \{s_{E,T}^i\}$ in the environment $E$ of task $T$, they are observed from two cameras and get two groups of observed images $\mathbb{O}_{1,E,T} = \{o_{1,E,T}^i\}$ and $\mathbb{O}_{2,E,T} = \{o_{2,E,T}^i\}$. We obtain the average pairwise distance between the latent representations of two visual encoders by calculating the mean of the pairwise distances across all input states:

$$\bar{d}_{cos} = \sum_{i=1}^{|\mathbb{S}_{E,T}|} \left(1 - S_C\left(g_1^E\left(o_{1,E,T}^i\right), g_2^E\left(o_{2,E,T}^i\right)\right)\right) / |\mathbb{S}_{E,T}|, \tag{11}$$

$$\bar{d}_{L2} = \sum_{i=1}^{|\mathbb{S}_{E,T}|} d_{L2}\left(g_1^E\left(o_{1,E,T}^i\right), g_2^E\left(o_{2,E,T}^i\right)\right) / |\mathbb{S}_{E,T}|, \tag{12}$$

where $g_1^E$ and $g_2^E$ are the visual encoders for $\mathbb{O}_{1,E,T}$ and $\mathbb{O}_{2,E,T}$ separately, $S_C(\boldsymbol{a}, \boldsymbol{b}) = \frac{\boldsymbol{ab}}{\|\boldsymbol{a}\|\|\boldsymbol{b}\|}$ is the cosine similarity and $d_{L2}(p,q) = \|p - q\|$ is the L2 distance. Fig. 8 shows the cosine and L2 distances in all the experiments in the Push task. We record the distances data and the mean cosine and L2 distances across all these experiments in Table 8.

## D    Influence of Using Decoder 1 and Decoder 2

This section aims to answer the question: Is there any difference between choosing which action decoder (action decoder 1 v.s. action decoder 2)?

We carry out the zero-shot transfer in the Stack task with six different visual configurations. For each experiment, we try action decoder 1 and action decoder 2, and report their success rates side-by-side in Table 9. We notice that the average success rates of decoder 1 and decoder 2 across the six visual configurations are close to each other for all the methods, and the maximum difference is within 15%. We believe that this difference is generated by the randomness of the testing process, but not the systematic advantage of one

|  | Cosine Distance | | L2 Distance | |
|---|---|---|---|---|
|  | ours | baseline | ours | baseline |
| Masked | **0.065** | 0.422 | **4.943** | 6.323 |
| Zoom in | **0.083** | 0.687 | **4.068** | 4.469 |
| Blurred | **0.043** | 0.982 | **4.937** | 5.828 |
| Gaussian Noise | **0.062** | 1.044 | **3.196** | 6.865 |
| Camera Type | **0.045** | 0.759 | **3.588** | 5.764 |
| Camera Position | **0.003** | 1.072 | **1.936** | 5.314 |
| Mean | **0.050** | 0.828 | **3.778** | 5.761 |

Table 8: **Latent Representations Distances Data.** The distances of the latent representations in all the experiments in the Push task are recorded. We also calculate the mean distances across all these experiments.

|  |  | Mask | Zoom in | Blurred | Noise | Fisheye | Camera Position | Average |
|---|---|---|---|---|---|---|---|---|
| Devin et al. 2017 | Decoder 1 | 0.7±0.94 | **8.0±1.63** | 0.7±0.94 | **24.0±2.83** | 0.0±0.00 | **14.0±3.27** | **7.9** |
|  | Decoder 2 | **5.3±2.49** | 0.0±0.00 | **14.0±2.83** | 6.7±2.49 | **3.3±1.89** | 5.3±2.49 | 5.8 |
| Cannistraci et al. 2024 (linear) | Decoder 1 | **47.3±0.94** | **62.0±4.32** | **32.7±3.77** | 30.7±0.94 | **54.0±8.64** | 14.7±6.18 | 40.2 |
|  | Decoder 2 | 39.3±4.11 | 56.7±1.89 | 26.7±6.60 | **51.3±4.11** | 46.0±2.83 | **23.3±1.89** | **40.6** |
| Cannistraci et al. 2024 (non-linear) | Decoder 1 | **10.0±1.63** | **12.0±0.00** | 0.0±0.00 | 3.3±0.94 | 0.0±0.00 | **0.7±0.94** | 4.3 |
|  | Decoder 2 | 5.3±0.94 | 6.7±0.94 | **8.0±2.83** | **14.7±1.89** | **2.0±1.63** | 0.0±0.00 | **6.1** |
| PeS (-w/o disent. loss) | Decoder 1 | **34.0±11.43** | 10.7±4.11 | **62.0±10.71** | **34.0±7.12** | 22.7±3.77 | **26.0±4.32** | 31.6 |
|  | Decoder 2 | 31.3±2.49 | **52.7±6.60** | 28.0±3.27 | 26.7±1.89 | **32.7±3.40** | 19.3±4.11 | **31.8** |
| PeS (w. l1 & l2 loss) | Decoder 1 | **92.7±0.94** | **98.0±0.00** | 62.7±6.60 | 24.0±4.90 | **59.3±7.36** | **58.7±1.88** | **65.9** |
|  | Decoder 2 | 72.7±3.77 | 36.0±1.63 | **88.7±4.99** | **58.7±1.88** | 23.3±3.40 | 33.3±0.94 | 52.1 |
| PeS | Decoder 1 | **94.7±0.94** | **96.7±0.94** | 90.0±1.63 | **96.7±1.89** | **97.3±2.49** | **80.0±4.90** | **92.6** |
|  | Decoder 2 | 83.3±4.11 | 86.0±4.32 | **94.0±2.83** | 92.7±0.94 | 95.3±0.94 | 68.7±6.18 | 85.0 |

Table 9: **Action Decoder 1 V.S. Action Decoder 2.** All the experiments are carried out with the Stack task. For the PeS method and other baseline and ablation methods, the difference in the average success rates of using action decoder 1 or action decoder 2 is within 15%. The choice of the action decoder does not have distinct influence on the zero-shot transfer success rates.

decoder to another decoder. This result suggests that the choice of the action decoder during the zero-shot transfer process does not make a distinct difference for all the methods on average.

If we look at the success rates of each visual configuration, we can see that the success rates difference of the PeS method is within 11%. The choice of decoder does not affect the reassembled policy's performance with PeS, and both decoders can lead to very satisfying success rates over 85%. However, in some experiments with the baselines and the ablation methods (e.g. PeS (w. l1 & l2 loss)-Fisheye), we can see a huge success rate difference. It indicates that the choice of different decoders has a random influence on the performance of the baselines and ablation methods for different visual configurations, but it doesn't affect the average success rates drastically.

## E   Anchors from failure trajectories

|  | Mask | Zoom in | Blurred | Noise | Fisheye | Camera Position | Average |
|---|---|---|---|---|---|---|---|
| 100% success rate | 94.7±0.94 | **96.7±0.94** | **90.0±1.63** | 96.7±1.89 | **97.3±2.49** | 80.0±4.90 | **92.6** |
| 54% success rate | **98.7±0.94** | 95.3±3.77 | 46.0±1.63 | 85.3±0.94 | 52.7±2.49 | 84.0±3.27 | 77.0 |
| 6% success rate | 49.3±6.80 | 48.7±4.11 | 76.7±8.06 | **98.7±0.94** | 44.7±3.40 | **98.0±1.63** | 69.4 |

Table 10: **Anchors from Failure Trajectories.** All the experiments are carried out with the Stack task. We chose anchor datasets from three different datasets with K-means algorithm: (1) The dataset is collected by an expert agent, and it contains 200 success trajectories. (2) The dataset is collected by an semi-trained policy which has around 50% of success rate. It is used to collect 200 trajectories, among which 54% are successful trajectories and 46% are failure trajectories. (3) The dataset is collected by a poorly behaved policy with a close to zero success rate. It is used to collect 200 trajectories, among which only 6% are successful trajectories and 94% are failure trajectories. The average success rate decreases when the percentage of successful trajectories in the dataset for anchor selection decreases.

This section aims to answer the question: Does the data collection process require some success trajectory? Or even failure/exploratory trajectories are still useful?

We use the Stack task to carry out the experiments. We collect a dataset with 200 successful trajectories and collect an anchor set 1 from this fully successful dataset. Then we train a policy to about 50% of training success rates and execute it to collect 200 trajectories during testing. 54% of these trajectories are successful and the rest 46% are failed. We collect an anchor set 2 from this semi-successful dataset. We also train a policy for only 1 minute so that it has a very low success rate. We use this poorly trained policy to collect a dataset of 200 trajectories with only 6% of them being successful. We collect an anchor set 3 from this dataset in which most trajectories fail. All the anchors are collected with the k-means algorithm as shown in Figure 3.

We use these three different anchor sets for training the policies across the six visual configurations. The data set used for training is still the same dataset collected by an expert agent with 200 successful trajectories, and the only difference is that the anchors are selected from data sets with different success rates. Table 10 reports the success rates. On average, the success rate decreases when the proportion of successful trajectories decreases for the anchor selection. In addition, we notice that the performance of the trained policy becomes unstable when the failure trajectory number during anchor selection increases. When there are no failure trajectories, the policy success rates are stably above 80%. In contrast, with the failure trajectories number increases, although there are still some cases where the trained policy can get over 90% of success rates, there appear more cases where the policy gets around 40% to 50% of success rates. These low-performance experiments make the average success rate decrease.

Since we need to use an expert dataset for training the policies, and it is a small data set with only 200 trajectories that are easy to collect, we encourage the users of the PeS method to directly collect the anchor set from the training data set with K-means algorithm. Our experiment result shows that this anchor selection method can lead to more stable zero-shot transfer performance and a higher average success rate.

## F   Latent Representations Visualization

We visualize the latent representations of the Lift, Can, Stack and Door tasks. Among all the visual configuration changing experiments, we choose the camera position variation experiments. We first reduced the 256D representations to 3D with PCA (Hotelling, 1933) for visualization. Since the side view encoder of the policy 2 is stitched to the policy 1 at the position of its original front view encoder, we compare the latent representations of these two encoders.

In these visualizations, the red data points are the first 5 steps of images in each of the 200 games in the dataset of a certain task. The blue data points are the last 5 steps of images, and the green data points are the 5 steps of images in the middle of each trajectory. Therefore, we have 1000 data points for each color which represent the starting, middle, and ending stages of a task.

The visualization results show that PeS can better align the latent space and force an approximate invariance of the latent representations. In contrast, the Devin et al. (2017) baseline without adopting the relative representation and the disentanglement regularization leads to different latent representations between the two encoders, and they have an approximately isometric transformation relationship. These visualizations support our conclusions in the paper.

## G   Impact of Anchor Number

To study the impact of number of anchors on transfer performance, we picked a challenging task, Stack, and reported the success rates with standard errors over 3 random seeds on all the different visual configurations in Table 11. We found that when the anchor number is too small, the latent space of the visual encoder does not have enough capacity to capture effective visual representations while maintaining an approximate invariance at the same time. Therefore, the zero-shot transfer performance drops drastically. On the other hand, when the number of anchors becomes too large, there are more redundant anchors which are similar to each other. This will make disentangling the features at the latent space with the disentanglement loss

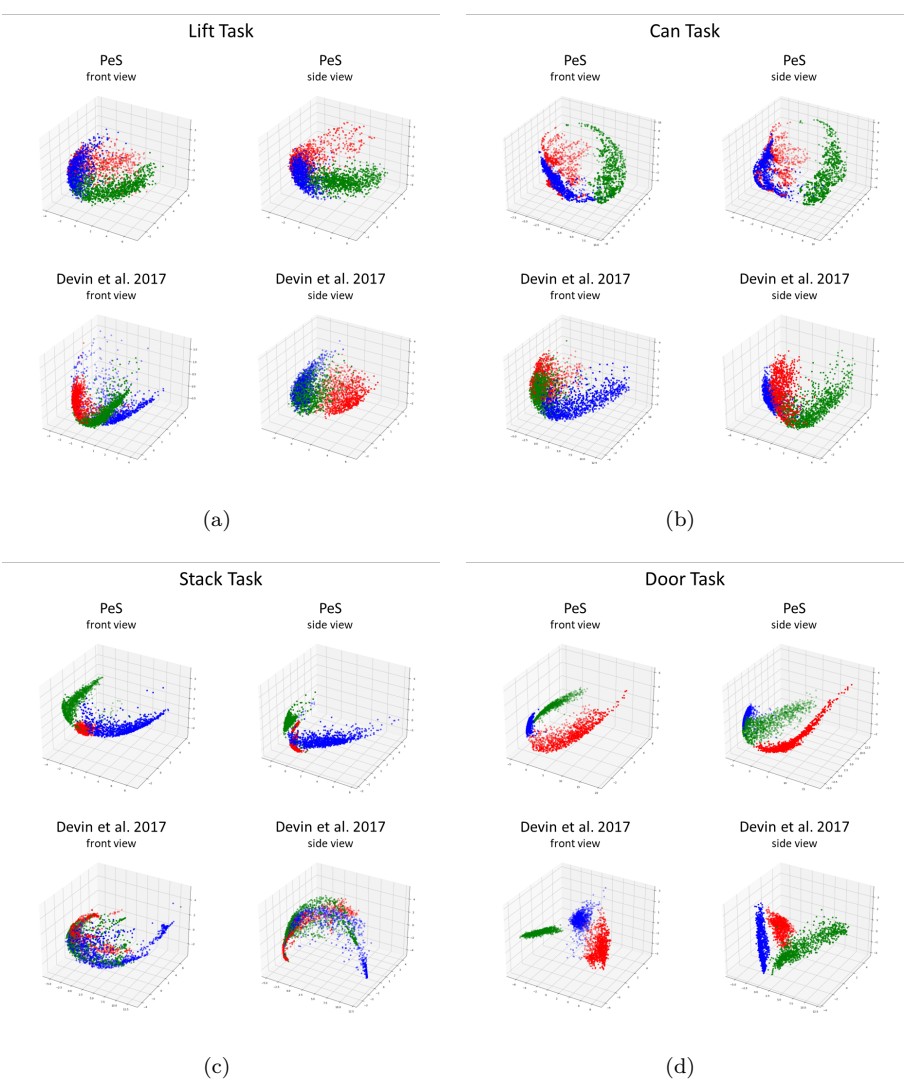

Figure 11: **Latent Representations Visualization.** We visualize the latent representations of the front-view encoder of policy 1 and the side-view encoder of policy 2 of the Lift, Can, Stack and Door tasks with the Camera Position changes in the visual configuration.

harder, which can also lead to some drop of the performance. We empirically found that an anchor number between 256 to 512 can achieve optimal performance in our tasks.

| Anchor Number | Masked | Zoom in | Blurred | Noise | Fisheye | Position | Average |
|---|---|---|---|---|---|---|---|
| 1024 | 72.7±0.94 | 56.0±4.32 | 88.7±3.40 | 86.7±3.77 | 65.3±2.49 | 69.3±2.49 | 73.1 |
| 512 | 90.7±3.40 | 94.7±0.94 | **91.3±0.94** | 92.7±5.73 | 95.3±3.40 | **88.7±9.84** | 92.2 |
| 256 | **94.7±0.94** | **96.7±0.94** | 90.0±1.63 | **96.7±1.89** | **97.3±2.49** | 80.0±4.90 | **92.6** |
| 128 | 72.7±10.87 | 43.3±5.73 | 42.0±1.63 | 24.0±2.83 | 73.3±1.89 | 30.0±7.12 | 47.6 |
| 64 | 1.3±0.00 | 6.7±0.94 | 6.7±4.11 | 0.0±0.00 | 5.3±2.49 | 0.0±0.00 | 3.3 |

Table 11: Success rates of the Stack task with different anchor numbers

We also investigated the impact of the anchor number on computational efficiency. We picked the Stack task and trained the policies on a NVIDIA A6000 GPU with the batch size of 32. As shown in Table 12, a larger anchor number will lead to a larger model size, longer anchor searching time, and longer training time for each epoch.

| Anchor Number | Model Size | Anchors Searching Time (s) | Training Time Per Epoch (s) |
|---|---|---|---|
| 1024 | 26,835,803 | 1455 | 182 |
| 512 | 25,720,667 | 922 | 107 |
| 256 | 25,163,099 | 570 | 66 |
| 128 | 24,884,315 | 288 | 33 |
| 64 | 24,744,923 | 194 | 23 |

Table 12: network model sizes, anchors searching time, and training time per epoch of the Stack task with different anchor numbers

To sum up, either a too large or too small anchor number can hurt the transfer performance, and larger anchor numbers can reduce the computational efficiency. It is important to select an appropriate anchor number for each task.

# H    Impact of Disentanglement Loss Weight

| Weights | Masked | Zoom in | Blurred | Noise | Fisheye | Camera Position | Average |
|---|---|---|---|---|---|---|---|
| 0.0 | 34.0±11.43 | 10.7±4.11 | 62.0±10.71 | 34.0±7.12 | 22.7±3.77 | 26.0±4.32 | 31.6 |
| 0.0002 | 34.7±1.89 | 8.0±1.63 | 88.0±2.83 | 30.0±1.63 | 16.0±2.83 | 24.7±2.49 | 33.6 |
| 0.002 | **94.7±0.94** | **96.7±0.94** | 90.0±1.63 | **96.7±1.89** | **97.3±2.49** | **80.0±4.90** | **92.6** |
| 0.02 | 90.7±2.49 | 80.7±10.87 | **92.0±4.90** | 88.0±3.27 | 81.3±5.25 | 70.0±4.32 | 83.8 |
| 0.2 | 35.3±3.40 | 56.0±1.63 | 88.7±2.49 | 31.3±0.05 | 48.0±1.63 | 16.0±4.32 | 45.9 |

Table 13: Success rates of the Stack task with different disentanglement loss weight

We have also studied the effect of different weights of disentanglement loss. We picked the Stack task and tested different weights on all the various visual configurations. When the weight is close to zero, the transfer performance gets close to that of the PeS (w/o disent. loss) ablation method and is significantly lower than the PeS full method. When the weight becomes too large, the optimization process leans too much to disentangling the latent features than imitating the expert behaviors. Therefore, the weaker imitation of the expert agent will also cause the performance drop. We empirically found that the optimal weight of the disentanglement loss in our tasks should be around the range of 0.002 to 0.02. The success rates with standard errors over 3 random seeds are shown in Table 13.

# I  Impact of Replay Trajectory Deviations

One limitation of PeS is that it requires a trajectory replay to collect the anchor images. However, in real-world applications, it is usually hard to accurately replay the exact trajectories. In order to understand how the trajectory deviation errors in replay could influence the performance of PeS, we conduct additional experiments to introduce trajectory deviations with different amplitudes.

We use the Stack task in the simulation to test the effect of trajectory deviations. During the trajectory replaying, we add a random horizontal vector to every position point on the trajectory except for the gripper close or open action point. The length of the random vector to cause deviation is sampled from a uniform distribution within a certain range, and we test out the range options of 0 to1 cm, 0 to 3 cm, and 0 to 5 cm. The direction of the deviation vector is randomly sampled within the x-y plane.

| Amplitude | Masked | Zoom in | Blurred | Noise | Fisheye | Position | Lighting | Average |
|---|---|---|---|---|---|---|---|---|
| 0 cm | 94.7±0.94 | 96.7±0.94 | 90.0±1.63 | 96.7±1.89 | 97.3±2.49 | 80.0±4.90 | 82.0±1.63 | 91.1 |
| 1 cm | 72.0±5.89 | 86.7±4.71 | 77.3±6.18 | 88.0±1.63 | 93.3±0.94 | 74.7±7.36 | 78.0±4.32 | 81.4 |
| 3 cm | 34.7±3.77 | 35.3±1.89 | 44.7±8.22 | 26.7±1.89 | 37.3±2.49 | 19.3±7.36 | 23.0±2.83 | 31.6 |
| 5 cm | 12.0±2.83 | 14.7±4.99 | 16.7±0.94 | 23.3±3.40 | 16.7±3.40 | 8.3±1.89 | 3.3±0.94 | 13.6 |

Table 14: Success rates of the Stack task with different replay trajectory deviation amplitudes

The experiment results are shown in the table below. When the trajectory deviation is within the range of 1 cm, The average performance of PeS drops by about 10%, but it can still achieve 81.4% of average success rate, which is much higher than all the baselines that don't require trajectory replay and all the ablation methods that require trajectory replay without deviation. When the trajectory deviation goes up to the range of 3cm, PeS can achieve an average success rate of 31.6%, which is on par with the best performing baselines RT1 (29.5%) and Cannistraci et al. 2024 (linear) (34.6%). When the trajectory deviation goes up to a very large range of 5cm, the average success rate of PeS drops to 13.6%.

In summary, although PeS currently requires trajectory replays for anchor selection, it doesn't require very precise replay and can perform well within 1 cm of replay error range.

