# OpenReview forum: "Perception Stitching: Zero-Shot Perception Encoder Transfer for Visuomotor Robot Policies"
_TMLR — Accepted by TMLR_

### Review · Reviewer_FoMS · 2024-08-26

**Summary Of Contributions:**

This paper introduces Perception Stitching (PeS), a novel approach for zero-shot transfer of visuomotor robot policies across different visual configurations. The key contributions include:

1. A modular visuomotor policy design that separates visual encoders and action decoders.
2. A method for latent space alignment using relative representations, enabling the reuse of visual encoders.
3. A novel anchor selection technique for imitation learning scenarios.
4. The application of disentanglement regularisation to improve latent space alignment.

The authors demonstrate the effectiveness of PeS through comprehensive experiments in both simulated and real-world environments, covering various manipulation tasks and visual configurations. The method consistently outperforms existing baselines, particularly in complex tasks and real-world scenarios.

**Audience:**

Yes

**Broader Impact Concerns:**

Given that the authors have already addressed these key points, there are no major omissions in their broader impact statement.

**Claims And Evidence:**

Yes

**Requested Changes:**

Critical changes:

1. Provide a more thorough discussion of potential limitations and failure cases of the proposed method.
2. While the authors compare with modular approaches (Devin et al. 2017 and Cannistraci et al. 2023), the paper could benefit from comparison or discussion with a recent end-to-end approach for zero-shot or few-shot transfer in visuomotor policies. Consider comparing with or discussing differences from an end-to-end method such as RT-1 (Brohan et al., 2022) or a similar recently proposed approach. This could highlight the trade-offs between modular and end-to-end architectures in the context of zero-shot transfer for robotic tasks.
3. Clarify the novelty of each component of PeS in relation to existing techniques, particularly the use of relative representations and disentanglement regularisation.
4. Provide an analysis and discussion of the method's sensitivity to hyperparameters, particularly the number of anchors and its impact on transfer performance and computational efficiency, and the weight of the disentanglement loss and its effect on the trade-off between transfer performance and representation quality.

Suggested improvements:

1. Expand real-world experiments to include a broader range of tasks and visual configurations, particularly those requiring long-horizon planning or multi-step manipulation sequences
2. While the paper does explore different camera configurations, consider investigating the method's robustness to more extreme domain shifts, such as significant lighting changes or novel object appearances.
3. Explore alternatives to k-means for anchor selection. How would density-based clustering methods or learnable anchor points affect performance and efficiency, especially in the presence of outliers.
4. Clarify why certain tasks may benefit from the temporal dependencies that RNNs can capture, while the other tasks may not require this temporal modeling.
5. Consider discussing whether data resampling methods that enhance the quality of behavioral cloning, such as Model-Based Trajectory Stitching (Hepburn & Montana, 2024), could potentially further improve the performance of your approach.

**Strengths And Weaknesses:**

Strengths:

1. Novel approach to zero-shot transfer in visuomotor policies.
2. Comprehensive evaluation in both simulated and real-world environments.
3. Convincing ablation studies and comparisons with relevant baselines.
4. In-depth analysis of latent representations and attention mechanisms using Grad-CAM.
5. Demonstrated effectiveness in complex manipulation tasks.
6. Potential for significant impact in robotics and transfer learning.

Weaknesses:

1. The method requires replaying trajectories to collect new datasets for anchor selection, which can be time-consuming in real-world settings.
2. Limited exploration of scalability to larger datasets or more diverse tasks.
3. Insufficient discussion of potential limitations or failure cases.
4. While the real-world experiments cover several tasks (Reach, Push, Lift, Stack), they are still limited in scope, particularly lacking long-horizon planning or multi-step manipulation sequences.

---

> ### Author Response · Authors · 2024-09-19
> **Response to Reviewer FoMS (part 1)**
>
> We thank the reviewer for taking the time and effort to review our manuscript and for their thoughtful comments. We are encouraged that you highlight our analysis of latent representations and find this work has potential for significant impact in robotics and transfer learning. We would like to address all your concerns and questions below with point responses:
>
> - **“The method requires replaying trajectories to collect new datasets for anchor selection, which can be time-consuming in real-world settings.”**
>
> We agree that our current setup requires replaying trajectories to collect new dataset for anchor selection. However, since we do not require human experts to demonstrate trajectories in the replaying process, we found that human efforts are minimum. In our future work, we believe that exploring better methods for anchor selection is an exciting research direction. We have also acknowledged this in our limitation section.
>
> - **“Limited exploration of scalability to larger datasets or more diverse tasks.”**
>
> Our experiments involve 5 simulation tasks and 4 real-world tasks. These tasks include basic manipulations such as “reach”, “push” and “lift”, as well as some difficult manipulations such as “can”, “stack”, and “door”. The difficult tasks involve multiple stages of precise manipulation, such as locating, grasping, lifting, accurately placing an object, and pulling an articulated object. In addition, we test each task in various visual configuration changes. Some changes are moderate such as Mask and Noise, while some others are very drastic such as Camera Position and Fisheye. Therefore, our current experiments have adequate diversity for testing the performance of PeS.
> We agree that exploring training on large-scale dataset is an interesting future direction. However, since the focus of our paper is not training large foundation models for robot learning, and typically hundreds to thousands of GPUs are required to train such models, we leave such exploration as future work. We have also acknowledged this in our paper.
>
> - **“While the real-world experiments cover several tasks (Reach, Push, Lift, Stack), they are still limited in scope, particularly lacking long-horizon planning or multi-step manipulation sequences.” & “Expand real-world experiments to include a broader range of tasks and visual configurations, particularly those requiring long-horizon planning or multi-step manipulation sequences.”**
>
> Some of our tasks such as stacking in the real world and opening door in the simulation has involved more complex manipulation tasks such as multi-step manipulation. Since this is our first step to explore representation alignment for perception stitching and transfer, we leave the exploration for tasks involving long-horizon planning in the future. We will acknowledge this in our revised paper.
>
> - **“Insufficient discussion of potential limitations or failure cases.” “Provide a more thorough discussion of potential limitations and failure cases of the proposed method.”**
>
> In addition to the reviewer’s points above that we have aknowledged, we will also include more discussions on limitations. For instance, we will discuss that PeS cannot yet achieve perfect performance on some tasks that require very high precision or have very long horizons. The first cause of this potential failure is that the Perception Stitching can only enforce an approximate invariance of the latent representations but not a strict invariance, which leads to the loss of some accuracy of the stitched policy. It is an interesting topic for the future work to further refine the latent space alignment. We will also discuss that the policy trained with the Behavior Cloning algorithm usually cannot recover from the error accumulation very well in a long horizon task. In the future work, we can possibly combine the idea of Perception Stitching with reinforcement learning or more advanced imitation learning that handles error accumulation or multimodal distributions, since these approaches have shown some capabilities of recovery from failures in recent works.

---

> > ### Author Response · Authors · 2024-09-19
> > **Response to Reviewer FoMS (part 5)**
> >
> > We also investigated the impact of the anchor number on computational efficiency.  We picked the Stack task and trained the policies on a NVIDIA A6000 GPU with the batch size of 32. As shown in the table below, a larger anchor number will lead to a larger model size, longer anchor searching time, and longer training time for each epoch.
> >
> > | Anchor Number   | Model Size   | Anchors Searching Time | Training Time Per Epoch |
> > | ------------ | ------------ | ---------------------- | ----------------------- |
> > | 1024         | 26835803     | 1455 second               | 182 second                 |
> > | 512          | 25720667     | 922 second                | 107 second                 |
> > | 256          | 25163099     | 570 second                | 66 second                  |
> > | 128          | 24884315     | 288 second                | 33 second                  |
> > | 64           | 24744923     | 194 second                | 23 second                  |
> >
> > To sum up, either a too large or too small anchor number can hurt the transfer performance, and larger anchor numbers can reduce the computational efficiency. It is important to select an appropriate anchor number for each task.
> >
> > In addition, we have studied the effect of different weights of disentanglement loss. We picked the Stack task and tested different weights on all the various visual configurations. When the weight is close to zero, the transfer performance gets close to that of the PeS (w/o disent. loss) ablation method and is significantly lower than the PeS full method. When the weight becomes too large, the optimization process leans too much to disentangling the latent features than imitating the expert behaviors. Therefore, the weaker imitation of the expert agent will also cause the performance drop. We empirically found that the optimal weight of the disentanglement loss in our tasks should be around the range of 0.002 to 0.02. The success rates with standard errors over 3 random seeds are shown below.
> > | Learning Rate | Masked       | Zoom in       | Blurred        | Noise         | Fisheye       | Camera Position   | Average |
> > | ------------- | ------------ | ------------- | -------------- | ------------- | ------------- | ----------------- | ------- |
> > | 0.0           | 34.0±11.43   | 10.7±4.11     | 62.0±10.71     | 34.0±7.12     | 22.7±3.77     | 26.0±4.32          | 31.6    |
> > | 0.0002        | 34.7±1.89    | 8.0±1.63      | 88.0±2.83      | 30.0±1.63     | 16.0±2.83     | 24.7±2.49          | 33.6    |
> > | 0.002         | **94.7±0.94**| **96.7±0.94** | 90.0±1.63      | **96.7±1.89** | **97.3±2.49** | **80.0±4.90**      | **92.6**|
> > | 0.02          | 90.7±2.49    | 80.7±10.87    | **92.0±4.90**  | 88.0±3.27     | 81.3±5.25     | 70.0±4.32          | 83.8    |
> > | 0.2           | 35.3±3.40    | 56.0±1.63     | 88.7±2.49      | 31.3±0.05     | 48.0±1.63     | 16.0±4.32          | 45.9    |
> >
> > We will include all the above experiments in our Appendix.
> >
> > - **“While the paper does explore different camera configurations, consider investigating the method's robustness to more extreme domain shifts, such as significant lighting changes or novel object appearances.”**
> >
> > We are glad to report that we have conducted additional experiments to investigate our method’s robustness to lighting changes. In these experiments, policy 1 is trained with a normal view 1 and a much brighter view 2 that mimics the overexposed situation. Policy 2 is trained with a much darker view 1 that mimics the underexposed situation and a normal view 2. Then we stitch the normal view 2 encoder from policy 2 to the normal view 1 encoder and action decoder of the policy 1. Our experiment results show that these drastic lighting changes can make the RT-1 baseline become almost malfunctional in the Can and Door task, while PeS can still achieve 73% of success rate in the Can task and 57% of success rate in the Door task. The success rates with standard errors over 3 random seeds are shown below, and we will add these results to our revised paper.

---

> > ### Comment · Reviewer_FoMS · 2024-10-07
> > **Thanks**
> >
> > Thank you for thoroughly addressing the questions raised in the review. While I understand that scalability can be addressed in future work, I still have concerns about the reliance on trajectory replays for anchor selection, which seems critical for real-world applications and was noted by other reviewers as well. I appreciate the robustness tests you’ve conducted under visual disturbances, such as lighting changes and noise, which demonstrate that your approach performs well under these conditions. However, would you consider performing additional simulations to introduce trajectory deviations—such as slight variations in movement paths or interaction dynamics? These tests could be conducted in a controlled environment to avoid the time-consuming nature of real-world experiments, while still mimicking real-world variability where exact trajectory replays may not be possible. I believe these additional tests could further support your claims and provide valuable insights into the method’s generalisation capabilities.

---

> > > ### Author Response · Authors · 2024-10-10
> > > **Replay Trajectory Deviations Experiments**
> > >
> > > Thank you for your positive feedback on our previous experiments to address your concerns. We agree that it is a great idea to perform additional simulations to introduce trajectory deviations, and we are glad to report that we have conducted experiments to explore how different level of variations in movement paths can affect the performance of PeS.
> > >
> > > We use the Stack task in the simulation to test the effect of trajectory deviations. During the trajectory replaying, we add a random horizontal vector to every position point on the trajectory except for the gripper close or open action point. The length of the random vector to cause deviation is sampled from a uniform distribution within a certain range, and we test out the range options of 0 to1 cm, 0 to 3 cm, and 0 to 5 cm. The direction of the deviation vector is randomly sampled within the x-y plane.
> > >
> > > The experiment results are shown in the table below. When the trajectory deviation is within the range of 1 cm, The average performance of PeS drops by about 10%, but it can still achieve 81.4% of average success rate, which is much higher than all the baselines that don’t require trajectory replay and all the ablation methods that require trajectory replay without deviation. When the trajectory deviation goes up to the range of 3cm, PeS can achieve an average success rate of 31.6%, which is on par with the best performing baselines RT1 (29.5%) and Cannistraci et al. 2024 (linear) (34.6%). When the trajectory deviation goes up to a very large range of 5cm, the average success rate of PeS drops to 13.6%.
> > >
> > > We would like to mention that most of the current robotic manipulation hardware is able to keep the positional error below 1cm, where PeS can achieve higher performance than other baselines as shown in our trajectory deviation experiments and our real-world experiments. The 3 cm or 5 cm deviation means that the deviated trajectory point is randomly sampled within a circle with 6 cm or 10 cm diameter around the real trajectory point. It is a very large error range, and it is reasonable to see the drop in performance of PeS.
> > >
> > > In summary, although PeS currently requires trajectory replays for anchor selection, it doesn’t require very precise replay and can perform well within 1 cm of replay error range. We believe these additional experiments further support our claims in this paper, and we thank the reviewer FoMS for proposing this insightful experiment.
> > >
> > >
> > >
> > > | Deviation Amplitude                       | Masked       | Zoom in      | Blurred      | Noise        | Fisheye       |Position      | Lighting      | Average |
> > > |----------------------------------|--------------|--------------|--------------|--------------|---------------|-----------------|---------------|---------|
> > > | 0 cm                             | 94.7±0.94    | 96.7±0.94    | 90.0±1.63    | 96.7±1.89    | 97.3±2.49     | 80.0±4.90       | 82.0±1.63     | 91.1    |
> > > | 1 cm                             | 72.0±5.89    | 86.7±4.71    | 77.3±6.18    | 88.0±1.63    | 93.3±0.94     | 74.7±7.36       | 78.0±4.32     | 81.4    |
> > > | 3 cm                             | 34.7±3.77    | 35.3±1.89    | 44.7±8.22    | 26.7±1.89    | 37.3±2.49     | 19.3±7.36       | 23.0±2.83     | 31.6    |
> > > | 5 cm                             | 12.0±2.83    | 14.7±4.99    | 16.7±0.94    | 23.3±3.40    | 16.7±3.40     | 8.3±1.89        | 3.3±-0.94     | 13.6    |

---

> > > ### Author Response · Authors · 2024-10-17
> > > **We would love to hear from you if you have further questions**
> > >
> > > Dear reviewer FoMS,
> > > Thank you again for your thoughtful review for our manuscript. We are following up to see if you have further questions about our paper. We aim to try our best to address your concerns of our paper. Thank you again!

---

> ### Author Response · Authors · 2024-09-19
> **Response to Reviewer FoMS (part 2)**
>
> - **“While the authors compare with modular approaches, the paper could benefit from comparison or discussion with a recent end-to-end approach for zero-shot or few-shot transfer in visuomotor policies. Consider comparing with or discussing differences from an end-to-end method such as RT-1 (Brohan et al., 2022) or a similar recently proposed approach.”**
>
> We are glad to report that we have added the experiments of comparing RT-1 for zero-shot transfer in visuomotor policies. We show the results as success rates with standard errors over 3 random seeds in the table below. The results show that the end-to-end method RT-1 demonstrates satisfying robustness on moderate visual variation such as “Mask” and “Noise”. For example, in the “Door-Mask”, “Door-Noise”, “Can-Noise” experiments, the RT-1 achieves higher success rates than our method (PeS). This is somewhat not surprising since RT-1 was trained on a much larger dataset with much larger model that could cover some of these variations in training. However, for the harder tasks with more drastic visual configuration changes, such as “Door-Fisheye” and “Stack-Camera Position”, we found that the RT-1 method could not perform well. These additional results suggest that the trade-offs between modular and end-to-end architectures is that end-to-end network is simpler in the end-to-end training procedure but usually cannot generalize to drastic changes in visual configuration that has not been encountered during training. In contrast, although the modular policies usually involve more careful structure designs in the training procedure, our results show that they performed better in drastic visual changes. We will add this result and analysis in the experiment section of our revised paper.
>
> - **Push:**
>
> | Method      | Mask  | Zoom in | Blurred | Noise | Fisheye | Position | Lighting | Average |
> | ------------- | ----- | ------- | ------- | ----- | ------- | -------- | -------- | ------- |
> | **RT1**       | 95.3±2.49 | 89.3±7.36 | 86.0±4.32 | 77.3±8.22 | 76.0±8.49 | 45.3±5.73 | 40.0±6.53 | 72.7 |
> | **Devin et al. 2017** | 60.7±10.6 | 8.7±4.99 | 16.7±3.77 | 59.3±6.80 | 29.3±7.36 | 19.3±5.73 | 36.7±3.40 | 33.0 |
> | **Cannistraci et al. 2024 (linear)** | 89.3±4.11 | 94.0±2.83 | 64.7±1.89 | 74.7±6.18 | 74.0±2.83 | 78.7±2.49 | 87.3±0.94 | 80.4 |
> | **Cannistraci et al. 2024 (non-linear)** | 12.7±1.89 | 18.7±4.99 | 42.8±3.27 | 23.3±0.94 | 6.0±4.32 | 5.3±2.49 | 32.7±3.40 | 20.2 |
> | **PeS (w/o disent. loss)** | **100.0±0.00** | 86.0±2.83 | 80.7±9.84 | **100.0±0.00** | **100.0±0.00** | **100.0±0.00** | 86.7±0.94 | 93.3 |
> | **PeS (w. l1 & l2 loss)** | 88.7±4.99 | 95.3±1.89 | 90.0±5.66 | **100.0±0.00** | 93.3±0.94 | 80.7±4.99 | 89.3±0.94 | 91.0 |
> | **PeS**        | **100.0±0.00** | **100.0±0.00** | **95.3±0.94** | **100.0±0.00** | 92.7±2.50 | **100.0±0.00** | **94.0±4.32** | **97.4** |
> - **Lift**
>
> | Method      | Mask  | Zoom in | Blurred | Noise | Fisheye | Position | Lighting | Average |
> | ------------- | ----- | ------- | ------- | ----- | ------- | -------- | -------- | ------- |
> | **RT1**       | 86.0±2.82 | 0.0±0.00 | 86.7±6.24 | 85.3±4.99 | 14.0±3.27 | 35.0±7.07 | 46.0±4.32 | 50.4 |
> | **Devin et al. 2017** | 0.0±0.00 | 5.3±2.49 | 48.0±5.89 | 9.3±4.11 | 14.7±4.99 | 36.0±1.63 | 18.7±4.99 | 18.9 |
> | **Cannistraci et al. 2024 (linear)** | 72.7±3.77 | 64.0±2.83 | 86.0±4.32 | 68.7±1.88 | **88.7±1.88** | 57.3±2.49 | 56.0±5.89 | 70.5 |
> | **Cannistraci et al. 2024 (non-linear)** | 89.3±2.49 | 36.0±3.27 | 52.7±3.40 | 93.3±2.49 | 16.7±2.49 | 21.3±0.94 | 37.3±2.49 | 49.5 |
> | **PeS (w/o disent. loss)** | 83.3±6.60 | 80.7±5.73 | 93.3±0.94 | 91.3±5.73 | 79.3±2.49 | **93.3±2.49** | 76.7±3.40 | 85.4 |
> | **PeS (w. l1 & l2 loss)** | **97.3±2.49** | 85.3±0.94 | **90.7±0.94** | 86.0±4.32 | 88.0±1.63 | 84.7±3.77 | 64.7±3.40 | 85.2 |
> | **PeS**        | 92.7±2.50 | **94.7±1.89** | 89.3±4.11 | **96.0±1.63** | **88.7±0.94** | 93.0±0.03 | **84.7±6.60** | **91.3** |

---

> ### Author Response · Authors · 2024-09-19
> **Response to Reviewer FoMS (part 3)**
>
> - **Can**
>
> | Method      | Mask  | Zoom in | Blurred | Noise | Fisheye | Position | Lighting | Average |
> | ------------- | ----- | ------- | ------- | ----- | ------- | -------- | -------- | ------- |
> | **RT1**       | **88.3±6.24** | 0.0±0.00 | **86.7±8.50** | **88.3±2.36** | 0.0±0.00 | 8.3±6.24 | 4.0±0.00 | 39.4 |
> | **Devin et al. 2017** | 19.3±5.25 | 24.7±1.89 | 2.67±1.89 | 6.0±4.32 | 29.3±3.40 | 1.3±1.89 | 3.3±1.89 | 12.4 |
> | **Cannistraci et al. 2024 (linear)** | 33.3±0.94 | 48.0±1.63 | 48.7±2.49 | 65.3±0.94 | 26.7±3.77 | 34.7±3.77 | 42.7±0.94 | 42.8 |
> | **Cannistraci et al. 2024 (non-linear)** | 72.7±0.94 | 24.7±2.49 | 37.3±4.99 | 42.7±3.40 | 8.7±1.89 | 39.3±1.89 | 38.7±1.89 | 37.7 |
> | **PeS (w/o disent. loss)** | 44.7±8.06 | 89.3±4.11 | 34.7±4.11 | 30.7±6.80 | **92.7±2.50** | 44.7±3.40 | 42.7±0.94 | 54.2 |
> | **PeS (w. l1 & l2 loss)** | 47.3±0.94 | 58.7±1.88 | 54.0±8.64 | 36.0±7.12 | 58.7±1.88 | 64.7±6.60 | 38.0±4.32 | 51.1 |
> | **PeS**        | 83.3±5.24 | **89.3±2.49** | 74.0±2.83 | 78.7±4.11 | 56.0±2.83 | **78.7±2.49** | **73.3±6.80** | **76.2** |
>
>
> - **Stack**
>
> | Method      | Mask  | Zoom in | Blurred | Noise | Fisheye | Position | Lighting | Average |
> | ------------- | ----- | ------- | ------- | ----- | ------- | -------- | -------- | ------- |
> | **RT1**       | 85.0±4.08 | 0.0±0.00 | 0.0±0.00 | 76.7±4.71 | 13.3±5.73 | 0.0±0.00 | 31.3±0.94 | 29.5 |
> | **Devin et al. 2017** | 0.7±0.94 | 8.0±1.63 | 0.7±0.94 | 24.0±2.83 | 0.0±0.00 | 14.0±3.27 | 4.7±2.49 | 7.4 |
> | **Cannistraci et al. 2024 (linear)** | 47.3±0.94 | 62.0±4.32 | 32.7±3.77 | 30.7±0.94 | 54.0±8.64 | 14.7±6.18 | 0.7±0.94 | 34.6 |
> | **Cannistraci et al. 2024 (non-linear)** | 10.0±1.63 | 12.0±0.00 | 0.0±0.00 | 3.3±0.94 | 0.0±0.00 | 0.7±0.94 | 8.7±3.77 | 5.0 |
> | **PeS (w/o disent. loss)** | 34.0±11.43 | 10.7±4.11 | 62.0±10.71 | 34.0±7.12 | 22.7±3.77 | 26.0±4.32 | 36.7±8.06 | 32.3 |
> | **PeS (w. l1 & l2 loss)** | 92.7±0.94 | **98.0±0.00** | 62.7±6.60 | 24.0±4.90 | 59.3±7.36 | 58.7±1.88 | 48.0±4.32 | 63.3 |
> | **PeS**        | **94.7±0.94** | 96.7±0.94 | **90.0±1.63** | **96.7±1.89** | **97.3±2.49** | **80.0±4.90** | **82.0±1.63** | **91.1** |
>
> - **Door**
>
> | Method      | Mask  | Zoom in | Blurred | Noise | Fisheye | Position | Lighting | Average |
> | ------------- | ----- | ------- | ------- | ----- | ------- | -------- | -------- | ------- |
> | **RT1**       | **96.0±1.63** | 0.0±0.00 | 26.7±3.40 | **94.7±3.40** | 0.7±0.94 | 0.0±0.00 | 0.0±0.00 | 31.15 |
> | **Devin et al. 2017** | 9.3±4.11 | 5.3±0.94 | 0.0±0.00 | 4.0±1.63 | 0.7±0.94 | 0.0±0.00 | 6.0±1.63 | 3.6 |
> | **Cannistraci et al. 2024 (linear)** | 0.0±0.00 | 1.3±0.94 | 10.7±2.49 | 10.7±4.99 | 2.0±1.63 | 47.3±9.29 | 12.0±7.12 | 12 |
> | **Cannistraci et al. 2024 (non-linear)** | 26.0±2.83 | 31.3±4.99 | 49.3±8.22 | 48.0±5.89 | 62.7±3.40 | 44.7±3.40 | 33.3±4.99 | 42.2 |
> | **PeS (w/o disent. loss)** | 24.7±7.71 | 44.0±2.83 | 34.7±3.77 | 0.7±0.94 | 36.7±0.94 | 23.3±3.40 | 24.0±2.83 | 26.9 |
> | **PeS (w. l1 & l2 loss)** | 4.0±1.63 | **78.0±5.66** | 3.3±0.94 | 2.0±1.63 | 42.7±4.99 | 6.0±3.26 | 14.7±4.99 | 21.5 |
> | **PeS**        | 58.7±4.11 | 68.7±0.94 | **70.7±0.94** | 52.7±3.40 | **64.7±4.99** | **48.7±3.40** | **56.7±5.73** | **60.1** |

---

> ### Author Response · Authors · 2024-09-19
> **Response to Reviewer FoMS ( part 4)**
>
> - **“Clarify the novelty of each component of PeS in relation to existing techniques, particularly the use of relative representations and disentanglement regularization.”**
>
> Our first novelty is our anchor selection technique that we have specifically designed for the robotic imitation learning scenario. The expert trajectories dataset does not have the labels of different classes that appear in the supervised learning scenario, as in the original relative representation paper, which means we cannot select representative anchors directly from different classes of the data. Therefore, we propose to adopt the K-means algorithm to divide the dataset into clusters and select the datapoints at the cluster centers as the anchors. We have contributed experiments to show that this method is effective for the first time. Moreover, we leverage the correspondence of the observation trajectories recorded by different cameras to collect corresponding anchors across different visual configurations. To the best of our knowledge, PeS is the first method that combines these techniques with relative representation to enable its strong performance in the imitation learning scenario.
>
> Our second novelty is our disentanglement regularization in PeS lies. Existing methods on relative representations do not include this component. Moreover, while previous works usually adopt disentailment regularization for abstracting visual representations in self-supervised learning [1] or unsupervised learning [2] for downstream computer vision tasks, to the best of our knowledge, PeS is the first method that shows the capability of disentanglement regularization to enhance the modularity of the neural networks. Our results show that adding this disentanglement have greatly improved our performance. Among all the related works of modular deep learning with relative representation, we believe PeS is the first method that discovers the performance advantage of combining relative representation with disentanglement regularization.
> Moreover, existing techniques have not been used in complex robot manipulation settings in our novel problem setup, perception stitching.
>
> [1] VICReg: Variance-Invariance-Covariance Regularization for Self-Supervised Learning
>
> [2] Are Disentangled Representations Helpful for Abstract Visual Reasoning?
>
> - **“Provide an analysis and discussion of the method's sensitivity to hyperparameters, particularly the number of anchors and its impact on transfer performance and computational efficiency, and the weight of the disentanglement loss and its effect on the trade-off between transfer performance and representation quality.”**
>
> We are happy to report that we have carried out experiments to study the impact of number of anchors on transfer performance and computational efficiency, and the weight of the disentanglement loss and its effect on the transfer performance.
>
> We picked a challenging task, Stack, and reported the success rates with standard errors over 3 random seeds on all the different visual configurations in the table below.  We found that when the anchor number is too small, the latent space of the visual encoder does not have enough capacity to capture effective visual representations while maintaining an approximate invariance at the same time. Therefore, the zero-shot transfer performance drops drastically. On the other hand, when the number of anchors becomes too large, there are more redundant anchors which are similar to each other. This will make disentangling the features at the latent space with the disentanglement loss harder, which can also lead to some drop of the performance. We empirically found that an anchor number between 256 to 512 can achieve optimal performance in our tasks.
>
> | Anchor Number   | Masked       | Zoom in       | Blurred        | Noise         | Fisheye       | Camera Position   | Average |
> | ------------ | ------------ | ------------- | -------------- | ------------- | ------------- | ----------------- | ------- |
> | 1024         | 72.7±0.94    | 56.0±4.32     | 88.7±3.40      | 86.7±3.77     | 65.3±2.49     | 69.3±2.49          | 73.1    |
> | 512          | 90.7±3.40    | 94.7±0.94     | **91.3±0.94**  | 92.7±5.73     | 95.3±3.40     | **88.7±9.84**      | 92.2    |
> | 256          | **94.7±0.94**| **96.7±0.94** | 90.0±1.63      | **96.7±1.89** | **97.3±2.49** | 80.0±4.90          | **92.6**|
> | 128          | 72.7±10.87   | 43.3±5.73     | 42.0±1.63      | 24.0±2.83     | 73.3±1.89     | 30.0±7.12          | 47.6    |
> | 64           | 1.3±0.00     | 6.7±0.94      | 6.7±4.11       | 0.0±0.00      | 5.3±2.49      | 0.0±0.00           | 3.3     |

---

> ### Author Response · Authors · 2024-09-19
> **Response to Reviewer FoMS (part 6)**
>
> - **Push:**
>
> | Method      | Mask  | Zoom in | Blurred | Noise | Fisheye | Position | Lighting | Average |
> | ------------- | ----- | ------- | ------- | ----- | ------- | -------- | -------- | ------- |
> | **RT1**       | 95.3±2.49 | 89.3±7.36 | 86.0±4.32 | 77.3±8.22 | 76.0±8.49 | 45.3±5.73 | 40.0±6.53 | 72.7 |
> | **Devin et al. 2017** | 60.7±10.6 | 8.7±4.99 | 16.7±3.77 | 59.3±6.80 | 29.3±7.36 | 19.3±5.73 | 36.7±3.40 | 33.0 |
> | **Cannistraci et al. 2024 (linear)** | 89.3±4.11 | 94.0±2.83 | 64.7±1.89 | 74.7±6.18 | 74.0±2.83 | 78.7±2.49 | 87.3±0.94 | 80.4 |
> | **Cannistraci et al. 2024 (non-linear)** | 12.7±1.89 | 18.7±4.99 | 42.8±3.27 | 23.3±0.94 | 6.0±4.32 | 5.3±2.49 | 32.7±3.40 | 20.2 |
> | **PeS (w/o disent. loss)** | **100.0±0.00** | 86.0±2.83 | 80.7±9.84 | **100.0±0.00** | **100.0±0.00** | **100.0±0.00** | 86.7±0.94 | 93.3 |
> | **PeS (w. l1 & l2 loss)** | 88.7±4.99 | 95.3±1.89 | 90.0±5.66 | **100.0±0.00** | 93.3±0.94 | 80.7±4.99 | 89.3±0.94 | 91.0 |
> | **PeS**        | **100.0±0.00** | **100.0±0.00** | **95.3±0.94** | **100.0±0.00** | 92.7±2.50 | **100.0±0.00** | **94.0±4.32** | **97.4** |
> - **Lift**
>
> | Method      | Mask  | Zoom in | Blurred | Noise | Fisheye | Position | Lighting | Average |
> | ------------- | ----- | ------- | ------- | ----- | ------- | -------- | -------- | ------- |
> | **RT1**       | 86.0±2.82 | 0.0±0.00 | 86.7±6.24 | 85.3±4.99 | 14.0±3.27 | 35.0±7.07 | 46.0±4.32 | 50.4 |
> | **Devin et al. 2017** | 0.0±0.00 | 5.3±2.49 | 48.0±5.89 | 9.3±4.11 | 14.7±4.99 | 36.0±1.63 | 18.7±4.99 | 18.9 |
> | **Cannistraci et al. 2024 (linear)** | 72.7±3.77 | 64.0±2.83 | 86.0±4.32 | 68.7±1.88 | **88.7±1.88** | 57.3±2.49 | 56.0±5.89 | 70.5 |
> | **Cannistraci et al. 2024 (non-linear)** | 89.3±2.49 | 36.0±3.27 | 52.7±3.40 | 93.3±2.49 | 16.7±2.49 | 21.3±0.94 | 37.3±2.49 | 49.5 |
> | **PeS (w/o disent. loss)** | 83.3±6.60 | 80.7±5.73 | 93.3±0.94 | 91.3±5.73 | 79.3±2.49 | **93.3±2.49** | 76.7±3.40 | 85.4 |
> | **PeS (w. l1 & l2 loss)** | **97.3±2.49** | 85.3±0.94 | **90.7±0.94** | 86.0±4.32 | 88.0±1.63 | 84.7±3.77 | 64.7±3.40 | 85.2 |
> | **PeS**        | 92.7±2.50 | **94.7±1.89** | 89.3±4.11 | **96.0±1.63** | **88.7±0.94** | 93.0±0.03 | **84.7±6.60** | **91.3** |
>
> - **Can**
>
> | Method      | Mask  | Zoom in | Blurred | Noise | Fisheye | Position | Lighting | Average |
> | ------------- | ----- | ------- | ------- | ----- | ------- | -------- | -------- | ------- |
> | **RT1**       | **88.3±6.24** | 0.0±0.00 | **86.7±8.50** | **88.3±2.36** | 0.0±0.00 | 8.3±6.24 | 4.0±0.00 | 39.4 |
> | **Devin et al. 2017** | 19.3±5.25 | 24.7±1.89 | 2.67±1.89 | 6.0±4.32 | 29.3±3.40 | 1.3±1.89 | 3.3±1.89 | 12.4 |
> | **Cannistraci et al. 2024 (linear)** | 33.3±0.94 | 48.0±1.63 | 48.7±2.49 | 65.3±0.94 | 26.7±3.77 | 34.7±3.77 | 42.7±0.94 | 42.8 |
> | **Cannistraci et al. 2024 (non-linear)** | 72.7±0.94 | 24.7±2.49 | 37.3±4.99 | 42.7±3.40 | 8.7±1.89 | 39.3±1.89 | 38.7±1.89 | 37.7 |
> | **PeS (w/o disent. loss)** | 44.7±8.06 | 89.3±4.11 | 34.7±4.11 | 30.7±6.80 | **92.7±2.50** | 44.7±3.40 | 42.7±0.94 | 54.2 |
> | **PeS (w. l1 & l2 loss)** | 47.3±0.94 | 58.7±1.88 | 54.0±8.64 | 36.0±7.12 | 58.7±1.88 | 64.7±6.60 | 38.0±4.32 | 51.1 |
> | **PeS**        | 83.3±5.24 | **89.3±2.49** | 74.0±2.83 | 78.7±4.11 | 56.0±2.83 | **78.7±2.49** | **73.3±6.80** | **76.2** |
>
>
> - **Stack**
>
> | Method      | Mask  | Zoom in | Blurred | Noise | Fisheye | Position | Lighting | Average |
> | ------------- | ----- | ------- | ------- | ----- | ------- | -------- | -------- | ------- |
> | **RT1**       | 85.0±4.08 | 0.0±0.00 | 0.0±0.00 | 76.7±4.71 | 13.3±5.73 | 0.0±0.00 | 31.3±0.94 | 29.5 |
> | **Devin et al. 2017** | 0.7±0.94 | 8.0±1.63 | 0.7±0.94 | 24.0±2.83 | 0.0±0.00 | 14.0±3.27 | 4.7±2.49 | 7.4 |
> | **Cannistraci et al. 2024 (linear)** | 47.3±0.94 | 62.0±4.32 | 32.7±3.77 | 30.7±0.94 | 54.0±8.64 | 14.7±6.18 | 0.7±0.94 | 34.6 |
> | **Cannistraci et al. 2024 (non-linear)** | 10.0±1.63 | 12.0±0.00 | 0.0±0.00 | 3.3±0.94 | 0.0±0.00 | 0.7±0.94 | 8.7±3.77 | 5.0 |
> | **PeS (w/o disent. loss)** | 34.0±11.43 | 10.7±4.11 | 62.0±10.71 | 34.0±7.12 | 22.7±3.77 | 26.0±4.32 | 36.7±8.06 | 32.3 |
> | **PeS (w. l1 & l2 loss)** | 92.7±0.94 | **98.0±0.00** | 62.7±6.60 | 24.0±4.90 | 59.3±7.36 | 58.7±1.88 | 48.0±4.32 | 63.3 |
> | **PeS**        | **94.7±0.94** | 96.7±0.94 | **90.0±1.63** | **96.7±1.89** | **97.3±2.49** | **80.0±4.90** | **82.0±1.63** | **91.1** |

---

> ### Author Response · Authors · 2024-09-19
> **Response to Reviewer FoMS (part 7)**
>
> - **Door**
>
> | Method      | Mask  | Zoom in | Blurred | Noise | Fisheye | Position | Lighting | Average |
> | ------------- | ----- | ------- | ------- | ----- | ------- | -------- | -------- | ------- |
> | **RT1**       | **96.0±1.63** | 0.0±0.00 | 26.7±3.40 | **94.7±3.40** | 0.7±0.94 | 0.0±0.00 | 0.0±0.00 | 31.15 |
> | **Devin et al. 2017** | 9.3±4.11 | 5.3±0.94 | 0.0±0.00 | 4.0±1.63 | 0.7±0.94 | 0.0±0.00 | 6.0±1.63 | 3.6 |
> | **Cannistraci et al. 2024 (linear)** | 0.0±0.00 | 1.3±0.94 | 10.7±2.49 | 10.7±4.99 | 2.0±1.63 | 47.3±9.29 | 12.0±7.12 | 12 |
> | **Cannistraci et al. 2024 (non-linear)** | 26.0±2.83 | 31.3±4.99 | 49.3±8.22 | 48.0±5.89 | 62.7±3.40 | 44.7±3.40 | 33.3±4.99 | 42.2 |
> | **PeS (w/o disent. loss)** | 24.7±7.71 | 44.0±2.83 | 34.7±3.77 | 0.7±0.94 | 36.7±0.94 | 23.3±3.40 | 24.0±2.83 | 26.9 |
> | **PeS (w. l1 & l2 loss)** | 4.0±1.63 | **78.0±5.66** | 3.3±0.94 | 2.0±1.63 | 42.7±4.99 | 6.0±3.26 | 14.7±4.99 | 21.5 |
> | **PeS**        | 58.7±4.11 | 68.7±0.94 | **70.7±0.94** | 52.7±3.40 | **64.7±4.99** | **48.7±3.40** | **56.7±5.73** | **60.1** |
>
> - **“Explore alternatives to k-means for anchor selection. How would density-based clustering methods or learnable anchor points affect performance and efficiency, especially in the presence of outliers.”**
>
> We are happy to share that we have conducted additional experiments to explore laternatives to k-means for anchor selection. Density-based clustering methods such as DBSCAN do not allow for direct control over the number of clusters, which is required for PeS to generate specific number of anchors near the clusters’ centers. Therefore, we explored to use the learning-based clustering method, Deep Embedded Clustering (DEC) [3], as the alternative to k-means for anchor selection. We picked the Stack task and tested to replace K-means with DEC in the PeS method. The results are shown in the last row, PeS (DEC anchors), of the table below. We found that replacing K-means with DEC does not have much effect on the transfer performance. However, considering that DEC requires longer training time to learn the anchor selection, we still recommend using the faster K-means algorithm. We will include this experiment results in the Appendix of our revised paper.
>
> | Category               | Mask       | Zoom in     | Blurred     | Noise       | Fisheye      | Position     | Lighting     | Average |
> | ---------------------- | ---------- | ----------- | ----------- | ----------- | ------------ | ------------ | ------------ | ------- |
> | **RT1**                | 85.0±4.08  | 0.0±0.00    | 0.0±0.00    | 76.7±4.71   | 13.3±5.73    | 0.0±0.00     | 31.3±0.94    | 29.5    |
> | **Devin et al. 2017**   | 0.7±0.94   | 8.0±1.63    | 0.7±0.94    | 24.0±2.83   | 0.0±0.00     | 14.0±3.27    | 4.7±2.49     | 7.4     |
> | **Cannistraci et al. 2024 (linear)**  | 47.3±0.94  | 62.0±4.32   | 32.7±3.77   | 30.7±0.94    | 54.0±8.64    | 14.7±6.18    | 0.7±0.94     | 34.6    |
> | **Cannistraci et al. 2024 (non-linear)** | 10.0±1.63 | 12.0±0.00 | 0.0±0.00   | 3.3±0.94     | 0.0±0.00     | 0.7±0.94     | 8.7±3.77     | 5       |
> | **PeS (w/o disent. loss)** | 34.0±11.43 | 10.7±4.11  | 62.0±10.71  | 34.0±7.12   | 22.7±3.77    | 26.0±4.32    | 36.7±8.06    | 32.3    |
> | **PeS (w. l1 & l2 loss)** | 92.7±0.94  | **98.0±0.00** | 62.7±6.60 | 24.0±4.90   | 59.3±7.36    | 58.7±1.88    | 48.0±4.32    | 63.3    |
> | **PeS**                | **94.7±0.94** | 96.7±0.94 | 90.0±1.63 | **96.7±1.89** | **97.3±2.49** | **80.0±4.90** | **82.0±1.63** | **91.1** |
> | **PeS (DEC anchors)**   | 92.7±0.94  | 86.0±1.63   | **92.0±1.63** | 79.3±3.40   | 87.3±2.49    | 75.3±0.94    | 77.3±4.71    | 84.3    |
>
> [3] Unsupervised Deep Embedding for Clustering Analysis

---

> ### Author Response · Authors · 2024-09-19
> **Response to Reviewer FoMS (part 8)**
>
> - **“Clarify why certain tasks may benefit from the temporal dependencies that RNNs can capture, while the other tasks may not require this temporal modeling.”**
>
> Certain tasks such as Can and Door may benefit from the temporal dependencies that RNNs can capture because these tasks involve multi-step manipulation in complicated visual backgrounds and occlusions. In these cases, we empirically found that capturing some historical information can help improve the success rates of the policies trained from scratch. On the other hand, for some simple tasks such as Push and Lift, they only involve single-step manipulation. In these cases, using an MLP action decoder is enough to train the policy to 100% success rate. When we experimented with replacing the MLP with RNN in these tasks, the RNN policies could also converge to 100% success rates, but they required more computational resources compared to the MLP policies. For the Stack task, although it is a multi-step task, we find that both MLP and RNN policies can converge to 100% success rate when training from scratch. We hypothesize that this is because the Stack task only involves two cubes and has a relatively simpler background with only a gray table, compared to the Can task with two different wooden tables and one of them has multiple compartments.
>
> - **“Consider discussing whether data resampling methods that enhance the quality of behavioral cloning, such as Model-Based Trajectory Stitching (Hepburn & Montana, 2024), could potentially further improve the performance of your approach.”**
>
> Model-Based Trajectory Stitching (TS) is a data resampling method that improves the quality of sub-optimal historical trajectories by ‘stitching’ together parts of the historical demonstrations to generate new, higher quality ones. By improving the data quality, we believe that it can definitely improve the performance of policies trained with behavior cloning (BC).
>
> For the current experiments in this paper, we make use of the high-quality expert data collected by proficient humans in the Robomimic [4] project. We used BC to train policies to 100% success rates first, and then stitch different network modules together. Since we train policies to 100% success rates with high-quality data and then do a zero-shot transfer, we found that our current setting is sufficient for our current experiments. However, we believe that methods such as Model-Based Trajectory Stitching can help improve the policy performance in more challenging tasks. We leave the exploration of using this method as future work. We will add this discussion of this paper to our revised paper.
>
> [4] What Matters in Learning from Offline Human Demonstrations for Robot Manipulation

---

### Review · Reviewer_rnZY · 2024-08-29

**Summary Of Contributions:**

The paper presents “Perception Stitching” (PeS), a novel approach that enables zero-shot transfer of visuomotor policies between environments with different visual configurations. The key contribution is the modularization and alignment of visual encoders via relative representations across policies to facilitate their reuse in novel perceptual configurations without retraining. The authors show that PeS disentangles perceptual knowledge from motion skills and demonstrates robust zero-shot transfer capabilities across different robotic tasks in both simulated and real-world environments. The method achieves higher success rates compared to baselines, especially in challenging tasks.

**Audience:**

Yes

**Broader Impact Concerns:**

The broader impact section is not present and not needed, according to my opinion.

**Claims And Evidence:**

Yes

**Requested Changes:**

*Sim2Real Data Collection:* It would be interesting to leverage on a sim2real approach where simulation environments are used to collect additional data for training or enhancing the real-world controller. This would allow the method to gather diverse data in a controlled simulation environment, which could then be transferred to the real controller, potentially reducing the reliance on real-world data collection and improving zero-shot transfer performance in real-world tasks.

*Explaining performance drops in real-world tests:* Some tasks in the real-world tests, especially Stack, exhibit a noticeable drop in performance compared to others like Reach or Push. Given that Stack involves multiple stages of precise manipulation—such as locating, grasping, lifting, and accurately placing an object—this complexity could naturally lead to more errors. However, the performance gap is significant, raising the question: Have you investigated the root causes behind this drop in performance?

*A reinforcement learning approach that could be relevant to your research:* While your work focuses on behavior cloning, have you considered looking into the following paper [1]? It explores the use of relative representations to achieve zero-shot stitching in simpler environments like CarRacing and the Atari suite. It may offer insights applicable to your approach, despite the difference in learning paradigms.

[1]: Zero-Shot Stitching in Reinforcement Learning using Relative Representations. https://arxiv.org/abs/2404.12917

**Strengths And Weaknesses:**

**Strengths**
1. *Method requires no training constraints*: Unlike many other methods that demand extensive pre-training on large datasets, fine-tuning on new tasks, or are limited to low-dimensional observations, PeS circumvents these constraints. By directly focusing on high-dimensional image observations, PeS can handle complex real-world robotic manipulation tasks. Additionally, PeS enables zero-shot transfer, meaning the system can generalize to novel visual configurations without the need for retraining or fine-tuning, which broadens its applicability to diverse environments and tasks without the overhead of additional training.
Moreover, the use of disentanglement regularization, ensures that the latent features are independent of each other, promoting better alignment between visual encoders.

2. *Strong results:* The experimental results clearly demonstrate that PeS consistently outperforms baseline methods, particularly in challenging tasks. Its ability to handle zero-shot transfer is especially notable in real-world tasks, where baseline methods struggle. In tasks like object manipulation or stacking, PeS achieves significantly higher success rates than competing approaches, showcasing its robustness and effectiveness in translating learned visuomotor policies from one context to another with minimal adaptation efforts.

3. *Comprehensive evaluation:* PeS was subjected to a wide array of evaluations, encompassing numerous simulated and real-world tasks. Tests include varying camera configurations, such as occlusions, zoom effects, and blurred or noisy images, number of cameras, ensuring that PeS was rigorously tested under challenging visual conditions. Additionally, the inclusion of real-world experiments provides a clear demonstration of how the method performs outside controlled simulations, which showcases its adaptability to various visuomotor tasks.

4. *Visual analysis with Grad-CAM:* The authors provide visual analysis of PeS’s performance through the use of Grad-CAM, highlighting which areas of the visual input the model focuses on when making decisions. The use of Grad-CAM allows for an intuitive understanding of why PeS succeeds where other methods fail, showing that PeS’s attention is directed to crucial task-relevant objects in the image.

**Weaknesses**

1. *Anchors collection:* The current requirement to replay trajectories for anchor selection imposes limitations on the method’s applicability, particularly in real-world settings where re-enacting the exact same trajectory can be time-consuming or impractical. A more flexible approach that enables automatic pairing of data across different visual encoders without needing identical trajectory replays could significantly broaden the method’s impact.

2. *Baseline Comparisons:* While PeS outperforms existing methods, the baselines used for comparison could be more diverse.

---

> ### Author Response · Authors · 2024-09-19
> **Response to Reviewer rnZY (part 1)**
>
> We thank the reviewer for taking the time and effort to review our manuscript and for the thoughtful comments. We appreciate your positive feedback on our novel approach, strong results and comprehensive evaluation and visual anslysis. We would like to address all of your concerns and questions below with point responses:
>
> - **“Anchors collection: The current requirement to replay trajectories for anchor selection imposes limitations on the method’s applicability, particularly in real-world settings where re-enacting the exact same trajectory can be time-consuming or impractical. A more flexible approach that enables automatic pairing of data across different visual encoders without needing identical trajectory replays could significantly broaden the method’s impact.”**
>
> We agree that automatic pairing data instead of requiring replaying trajectories can help improve our paper.  We have also acknowledged this in our limitation section. In our current setup, we have carried out multiple real-world experiments ranging from simple tasks to difficult tasks to demonstrate that this re-enacting is still practical in diverse real-world applications. Since we do not require human expert demonstrations during this replaying process, the human involvement is minimum. We feel very excited about the idea of automatic pairing of data across different visual encoders without trajectory replays, and we leave the exploration in this direction to future work.
>
> - **“While PeS outperforms existing methods, the baselines used for comparison could be more diverse.”**
>
> We are glad to report that we have added the experiments of comparing with RT-1 [1] as another baseline for zero-shot transfer in visuomotor policies. The previous baseline approaches (Devin et al. 2017 and Cannistraci et al. 2023) all adopt modular network architectures. In contrast, RT-1 adopts an end-to-end transformer architecture with training on a much larger and diverse dataset. RT-1 has also been reported to also have some extent of zero-shot transfer capability. We believe that adding this RT-1 baseline can make the comparison more diverse.
>
> We show the results as success rates with standard errors over 3 random seeds in the table below. The results show that the end-to-end method RT-1 demonstrates satisfying robustness on moderate visual variation such as “Mask” and “Noise”. For example, in the “Door-Mask”, “Door-Noise”, “Can-Noise” experiments, the RT-1 achieves higher success rates than our method (PeS). This is somewhat not surprising since RT-1 was trained on much larger dataset with much larger model that could cover some of these variations in training. However, for the harder tasks with more drastic visual configuration changes, such as “Door-Fisheye” and “Stack-Camera Position”, we found that the RT-1 method could not perform well. These additional results suggest that the trade-offs between modular and end-to-end architectures is that end-to-end network is simpler in the end-to-end training procedure but usually cannot generalize to drastic changes in visual configuration that has not been encountered during training. In contrast, although the modular policies usually involve more careful structure designs in the training procedure, our results show that they performed better in drastic visual changes. We will add this result and analysis in the experiment section of our revised paper.
>
> - **Push:**
>
> | Method      | Mask  | Zoom in | Blurred | Noise | Fisheye | Position | Lighting | Average |
> | ------------- | ----- | ------- | ------- | ----- | ------- | -------- | -------- | ------- |
> | **RT1**       | 95.3±2.49 | 89.3±7.36 | 86.0±4.32 | 77.3±8.22 | 76.0±8.49 | 45.3±5.73 | 40.0±6.53 | 72.7 |
> | **Devin et al. 2017** | 60.7±10.6 | 8.7±4.99 | 16.7±3.77 | 59.3±6.80 | 29.3±7.36 | 19.3±5.73 | 36.7±3.40 | 33.0 |
> | **Cannistraci et al. 2024 (linear)** | 89.3±4.11 | 94.0±2.83 | 64.7±1.89 | 74.7±6.18 | 74.0±2.83 | 78.7±2.49 | 87.3±0.94 | 80.4 |
> | **Cannistraci et al. 2024 (non-linear)** | 12.7±1.89 | 18.7±4.99 | 42.8±3.27 | 23.3±0.94 | 6.0±4.32 | 5.3±2.49 | 32.7±3.40 | 20.2 |
> | **PeS (w/o disent. loss)** | **100.0±0.00** | 86.0±2.83 | 80.7±9.84 | **100.0±0.00** | **100.0±0.00** | **100.0±0.00** | 86.7±0.94 | 93.3 |
> | **PeS (w. l1 & l2 loss)** | 88.7±4.99 | 95.3±1.89 | 90.0±5.66 | **100.0±0.00** | 93.3±0.94 | 80.7±4.99 | 89.3±0.94 | 91.0 |
> | **PeS**        | **100.0±0.00** | **100.0±0.00** | **95.3±0.94** | **100.0±0.00** | 92.7±2.50 | **100.0±0.00** | **94.0±4.32** | **97.4** |

---

> ### Author Response · Authors · 2024-09-19
> **Response to Reviewer rnZY (part 2)**
>
> - **Lift**
>
> | Method      | Mask  | Zoom in | Blurred | Noise | Fisheye | Position | Lighting | Average |
> | ------------- | ----- | ------- | ------- | ----- | ------- | -------- | -------- | ------- |
> | **RT1**       | 86.0±2.82 | 0.0±0.00 | 86.7±6.24 | 85.3±4.99 | 14.0±3.27 | 35.0±7.07 | 46.0±4.32 | 50.4 |
> | **Devin et al. 2017** | 0.0±0.00 | 5.3±2.49 | 48.0±5.89 | 9.3±4.11 | 14.7±4.99 | 36.0±1.63 | 18.7±4.99 | 18.9 |
> | **Cannistraci et al. 2024 (linear)** | 72.7±3.77 | 64.0±2.83 | 86.0±4.32 | 68.7±1.88 | **88.7±1.88** | 57.3±2.49 | 56.0±5.89 | 70.5 |
> | **Cannistraci et al. 2024 (non-linear)** | 89.3±2.49 | 36.0±3.27 | 52.7±3.40 | 93.3±2.49 | 16.7±2.49 | 21.3±0.94 | 37.3±2.49 | 49.5 |
> | **PeS (w/o disent. loss)** | 83.3±6.60 | 80.7±5.73 | 93.3±0.94 | 91.3±5.73 | 79.3±2.49 | **93.3±2.49** | 76.7±3.40 | 85.4 |
> | **PeS (w. l1 & l2 loss)** | **97.3±2.49** | 85.3±0.94 | **90.7±0.94** | 86.0±4.32 | 88.0±1.63 | 84.7±3.77 | 64.7±3.40 | 85.2 |
> | **PeS**        | 92.7±2.50 | **94.7±1.89** | 89.3±4.11 | **96.0±1.63** | **88.7±0.94** | 93.0±0.03 | **84.7±6.60** | **91.3** |
>
> - **Can**
>
> | Method      | Mask  | Zoom in | Blurred | Noise | Fisheye | Position | Lighting | Average |
> | ------------- | ----- | ------- | ------- | ----- | ------- | -------- | -------- | ------- |
> | **RT1**       | **88.3±6.24** | 0.0±0.00 | **86.7±8.50** | **88.3±2.36** | 0.0±0.00 | 8.3±6.24 | 4.0±0.00 | 39.4 |
> | **Devin et al. 2017** | 19.3±5.25 | 24.7±1.89 | 2.67±1.89 | 6.0±4.32 | 29.3±3.40 | 1.3±1.89 | 3.3±1.89 | 12.4 |
> | **Cannistraci et al. 2024 (linear)** | 33.3±0.94 | 48.0±1.63 | 48.7±2.49 | 65.3±0.94 | 26.7±3.77 | 34.7±3.77 | 42.7±0.94 | 42.8 |
> | **Cannistraci et al. 2024 (non-linear)** | 72.7±0.94 | 24.7±2.49 | 37.3±4.99 | 42.7±3.40 | 8.7±1.89 | 39.3±1.89 | 38.7±1.89 | 37.7 |
> | **PeS (w/o disent. loss)** | 44.7±8.06 | 89.3±4.11 | 34.7±4.11 | 30.7±6.80 | **92.7±2.50** | 44.7±3.40 | 42.7±0.94 | 54.2 |
> | **PeS (w. l1 & l2 loss)** | 47.3±0.94 | 58.7±1.88 | 54.0±8.64 | 36.0±7.12 | 58.7±1.88 | 64.7±6.60 | 38.0±4.32 | 51.1 |
> | **PeS**        | 83.3±5.24 | **89.3±2.49** | 74.0±2.83 | 78.7±4.11 | 56.0±2.83 | **78.7±2.49** | **73.3±6.80** | **76.2** |
>
>
> - **Stack**
>
> | Method      | Mask  | Zoom in | Blurred | Noise | Fisheye | Position | Lighting | Average |
> | ------------- | ----- | ------- | ------- | ----- | ------- | -------- | -------- | ------- |
> | **RT1**       | 85.0±4.08 | 0.0±0.00 | 0.0±0.00 | 76.7±4.71 | 13.3±5.73 | 0.0±0.00 | 31.3±0.94 | 29.5 |
> | **Devin et al. 2017** | 0.7±0.94 | 8.0±1.63 | 0.7±0.94 | 24.0±2.83 | 0.0±0.00 | 14.0±3.27 | 4.7±2.49 | 7.4 |
> | **Cannistraci et al. 2024 (linear)** | 47.3±0.94 | 62.0±4.32 | 32.7±3.77 | 30.7±0.94 | 54.0±8.64 | 14.7±6.18 | 0.7±0.94 | 34.6 |
> | **Cannistraci et al. 2024 (non-linear)** | 10.0±1.63 | 12.0±0.00 | 0.0±0.00 | 3.3±0.94 | 0.0±0.00 | 0.7±0.94 | 8.7±3.77 | 5.0 |
> | **PeS (w/o disent. loss)** | 34.0±11.43 | 10.7±4.11 | 62.0±10.71 | 34.0±7.12 | 22.7±3.77 | 26.0±4.32 | 36.7±8.06 | 32.3 |
> | **PeS (w. l1 & l2 loss)** | 92.7±0.94 | **98.0±0.00** | 62.7±6.60 | 24.0±4.90 | 59.3±7.36 | 58.7±1.88 | 48.0±4.32 | 63.3 |
> | **PeS**        | **94.7±0.94** | 96.7±0.94 | **90.0±1.63** | **96.7±1.89** | **97.3±2.49** | **80.0±4.90** | **82.0±1.63** | **91.1** |
>
> - **Door**
>
> | Method      | Mask  | Zoom in | Blurred | Noise | Fisheye | Position | Lighting | Average |
> | ------------- | ----- | ------- | ------- | ----- | ------- | -------- | -------- | ------- |
> | **RT1**       | **96.0±1.63** | 0.0±0.00 | 26.7±3.40 | **94.7±3.40** | 0.7±0.94 | 0.0±0.00 | 0.0±0.00 | 31.15 |
> | **Devin et al. 2017** | 9.3±4.11 | 5.3±0.94 | 0.0±0.00 | 4.0±1.63 | 0.7±0.94 | 0.0±0.00 | 6.0±1.63 | 3.6 |
> | **Cannistraci et al. 2024 (linear)** | 0.0±0.00 | 1.3±0.94 | 10.7±2.49 | 10.7±4.99 | 2.0±1.63 | 47.3±9.29 | 12.0±7.12 | 12 |
> | **Cannistraci et al. 2024 (non-linear)** | 26.0±2.83 | 31.3±4.99 | 49.3±8.22 | 48.0±5.89 | 62.7±3.40 | 44.7±3.40 | 33.3±4.99 | 42.2 |
> | **PeS (w/o disent. loss)** | 24.7±7.71 | 44.0±2.83 | 34.7±3.77 | 0.7±0.94 | 36.7±0.94 | 23.3±3.40 | 24.0±2.83 | 26.9 |
> | **PeS (w. l1 & l2 loss)** | 4.0±1.63 | **78.0±5.66** | 3.3±0.94 | 2.0±1.63 | 42.7±4.99 | 6.0±3.26 | 14.7±4.99 | 21.5 |
> | **PeS**        | 58.7±4.11 | 68.7±0.94 | **70.7±0.94** | 52.7±3.40 | **64.7±4.99** | **48.7±3.40** | **56.7±5.73** | **60.1** |
>
> [1] RT-1: Robotics Transformer for Real-World Control at Scale

---

> ### Author Response · Authors · 2024-09-19
> **Response to Reviewer rnZY (part 3)**
>
> - **“Sim2Real Data Collection: It would be interesting to leverage on a sim2real approach where simulation environments are used to collect additional data for training or enhancing the real-world controller. This would allow the method to gather diverse data in a controlled simulation environment, which could then be transferred to the real controller, potentially reducing the reliance on real-world data collection and improving zero-shot transfer performance in real-world tasks.”**
>
> We agree that exploring the usage of simulation to generate more data  can be used to transfer to the real controller and reduce the reliance on real-world training or expert data collection. In our current setting, we realize that sim2real is an orthogonal component that we can introduce to strengthen the overall deployment to real-world systems, since our method, Perception Stitching, mainly focuses on learning reusable policy modules, which has been shown to be effective in our experiments. We leave the integration of sim2real data collection as future work. We expect this will be particularly useful for more complex tasks that cannot be easily trained with limited real-world data yet.
>
> - **“Explaining performance drops in real-world tests: Some tasks in the real-world tests, especially Stack, exhibit a noticeable drop in performance compared to others like Reach or Push. Given that Stack involves multiple stages of precise manipulation—such as locating, grasping, lifting, and accurately placing an object—this complexity could naturally lead to more errors. However, the performance gap is significant, raising the question: Have you investigated the root causes behind this drop in performance?”**
>
> Yes, we have investigated the causes behind this drop in performance. For the real-world Stack task, most of the failure cases can be categorized into two classes. The first class of failures is that the robot fails to locate the exact position of the cube. In this situation, the robot usually reaches some location near the object but is not accurate enough to grasp the object. The second class is that the robot places the cube offset from the top of another cube, which causes the cube to be unstable on the other cube and slide off.
>
> We hypothesize that the first class of failures is mainly caused by the complex lighting condition in the real world compared to that in the simulation. The cube being observed from a different angle usually has a different reflection of light not recorded in the dataset, which makes the visuomotor policy fail to accurately locate the cube position. The second class of failures is not only caused by the complex lighting reflection of the bottom cube, but also caused by the accumulated error in a long horizon of manipulation. Since the action command includes the displacement of the end effector position, the hardware displacement errors will accumulate throughout the entire trajectory, which finally leads to the drift of the placement position on top of the cube. Then this offset placement causes the cube to slide off from the top of another cube.
>
> To sum up, the complex lighting reflection and the accumulation of hardware errors might be the main causes of the drop in performance. Collecting a larger dataset that includes more diverse lighting conditions and incorporating more advanced imitation learning that considers error accumulation could possibly improve the real-world performance.
>
> - **“A reinforcement learning approach that could be relevant to your research: While your work focuses on behavior cloning, have you considered looking into the following paper (Zero-Shot Stitching in Reinforcement Learning using Relative Representations)? It explores the use of relative representations to achieve zero-shot stitching in simpler environments like CarRacing and the Atari suite. It may offer insights applicable to your approach, despite the difference in learning paradigms.”**
>
> We thank the reviewer for pointing out this related work. We agree that, although our paper focuses on imitation learning, one exciting future direction is to explore our techniques in a reinforcement learning setup. We believe that the insights from this paper can help with this future exploration. We will include this discussion and the above paper in our future work discussion in the revised paper.

---

> > ### Comment · Reviewer_rnZY · 2024-10-09
> > **Satisfying response, I recommend acceptance**
> >
> > Thanks for the additional tests! Comparing your approach to RT-1 strengthens your method’s appeal, as it highlights how your modular approach outperforms transformer-based architectures when dealing with more drastic visual changes. This comparison emphasizes the adaptability and robustness of your technique in scenarios where visual conditions are more challenging.
> >
> > Your investigation into the failure cases is thorough and offers valuable insights into the limitations and potential areas for improvement.
> >
> > I am highly satisfied with both the content of the paper and the responses provided by the authors after the reviews. This work represents an interesting and innovative approach to modularity in reinforcement learning and neural networks more broadly. As a research paper, it is expected that there are some limitations, but these are acknowledged and can be further explored in future research. This includes the real-world replaying of trajectories, which, although a potential bottleneck, has been addressed in the paper without raising major concerns. I also appreciate your clarification that some requests, are orthogonal to your current focus and can also be explored in future work.
> >
> > I recommend the acceptance of this work.

---

> > > ### Author Response · Authors · 2024-10-10
> > > **Thank you**
> > >
> > > We appreciate the reviewer for the great support and the recognition of our work. Thank you!

---

### Review · Reviewer_p1mS · 2024-09-20

**Summary Of Contributions:**

The paper proposes a simple zero-shot approach to stitch the visual encoders of a robot policy with the visual encoders and decoder of another one. The method requires collecting a correspondent dataset of anchor images, and using a relative representation which projects the latents from the encoders to the anchor bases. The other important proposed improvement is the addition of batch covariance regularization of the encoder latents to the behavior cloning loss of policies. The stitching happens in a straightforward way, by projecting the new encoder to the bases of the correspondent anchor vectors. The paper evaluates the method compared to baselines across several simulated and real world robotics tasks, with many different camera configurations.

**Audience:**

Yes

**Broader Impact Concerns:**

No broader impact concern.

**Claims And Evidence:**

Yes

**Requested Changes:**

1. An important comparison that seems to be missing is what the performance of policies 1 and 2 to be stitched are in the first place. Only the performance of stitched policies are shown for the experiments. I would assume policy stitching even when done well would generally result in some degradation of performance from the base policies unless one of the base policy decoders isn’t good for some reason, or the other camera configuration for that policy is faulty. If policy 2 is worse than policy 1 (suppose it is trained with less data), then would stitching a policy 2 observation to policy 1 decoder result in improved performance? This would be of interest for example if policy 1 is some pretrained open model available, and policy 2 can be cheaply trained in a new setting with a different camera configuration. It would be very useful if the authors can provide performance of the base policy 1 and policy 2 in all their experiments.
2. I dont fully understand the single camera view experiment. If you train the two policies such that one of the encoders contains no information because of the completely dark input, won’t the decoder learn to completely just ignore the latents from that encoder? So for zero shot stitching where you use the policy 1 decoder, wouldn't you expect to get essentially the same performance as policy 1 before stitching? What was the performance of the policies before and after stitching? I would appreciate this section having a bit more clarification.
3. Given that the main contribution of this work is dependent on this alignment of latents from different views after the relative representation step, it would be nice if latent visualization from other tasks could also be added to the main paper or atleast the appendix.

**Strengths And Weaknesses:**

## Strengths
1. The paper presents well and is easy to understand.
2. The wide variety of experimental configurations in simulation and real world setup are well appreciated.
3. The visualization of latent correspondence in Figure 8 and cosine distance of embeddings is reasonable evidence as to the claims for why the proposed approach outperforms baselines. Please also see related requested change 3.
4. Covariance regularization is a simple to implement but very useful contribution here compared to previous approaches for this task.

## Weaknesses
1. The motivation is pitched as solving the problem of different institutions having varying robotics setups, and transferring policies can be difficult for example when camera configurations change. However, as noted by the authors in the limitations, replaying trajectories exactly to create duplicate anchors dataset in the new environment E_2 would be a severe bottleneck in many real world domains where controlling the environment variables to behave identically is difficult. Collecting a new dataset here and just training with BC might be preferable to doing stitching, which diminishes the value of this method.
2. Can this stitching actually improve over the policy trained with little data that can be collected in the new setting directly? If there is a strong pretrained policy trained in one setting, and is being transferred to a new setting with a different camera configuration, how much does this transfer preserve the performance of the policy in the initial setting? Please see the requested change 1. below where I follow up on this.

---

> ### Author Response · Authors · 2024-09-30
> **Response to Reviewer p1mS (part 1)**
>
> We thank the reviewer for taking the time and effort to review our manuscript and for the thoughtful comments. We appreciate your positive feedback on our presentation, abundant experiments, and visualization analysis. We are encouraged that you have highlighted the contribution of the covariance regularization in our proposed method. We would like to address all your concerns and questions below with point responses:
>
> -  **“The motivation is pitched as solving the problem of different institutions having varying robotics setups, and transferring policies can be difficult for example when camera configurations change. However, as noted by the authors in the limitations, replaying trajectories exactly to create duplicate anchors dataset in the new environment E_2 would be a severe bottleneck in many real world domains where controlling the environment variables to behave identically is difficult. Collecting a new dataset here and just training with BC might be preferable to doing stitching, which diminishes the value of this method.”**
>
> Our current setup indeed requires replaying trajectories to collect new dataset for anchor selection. However, since we do not require human experts to demonstrate trajectories in the replaying process, in our experiments, we found that human efforts are minimum. In contrast, collecting a new dataset and then training with BC usually requires a human to demonstrate the required number of expert trajectories again with full human supervision.
>
> In addition, we would like to clarify that the base policy 1 and 2 are trained in E_1 and E_2 separately, and then the reassembled policy is zero-shot transferred to a new environment E_3 with a new combination of visual configurations which has not been seen in the previous training. With PeS, the policy can achieve higher success rates compared to other baselines without the need of recollecting a new dataset and training with BC in E_3. On contrary, conventional imitation learning with BC will require collecting a new dataset and training from scratch in E_3, which is much more time consuming than PeS.
>
> Moreover, our real-world experiments have shown that PeS doesn’t require the environment variables to be strictly identical but only needs them to be similar. For example, in the real-world Lift task the grasp positions of the cube during the trajectory replaying are not exactly the same as that in the original trajectories, but PeS can still achieve 80% of success rate. We believe that our real-world experiments have proven that PeS can be successfully applied in the real world.
>
> In summary, PeS requires an automatic trajectory replay in E_2 and then can zero-shot transfer to E_3, which consumes less time and human efforts than collecting data and training from scratch in E_3 with BC. PeS asks the environment variables to be similar for the trajectory replay but does not require them to be strictly identical, and we have shown its capability in various real-world experiments. In our future work, we believe that exploring better methods for anchor selection is an exciting research direction. We have also acknowledged this in our limitation section.
>
> - **“Can this stitching actually improve over the policy trained with little data that can be collected in the new setting directly? If there is a strong pretrained policy trained in one setting, and is being transferred to a new setting with a different camera configuration, how much does this transfer preserve the performance of the policy in the initial setting? An important comparison that seems to be missing is what the performance of policies 1 and 2 to be stitched are in the first place. Only the performance of stitched policies are shown for the experiments. I would assume policy stitching even when done well would generally result in some degradation of performance from the base policies unless one of the base policy decoders isn’t good for some reason, or the other camera configuration for that policy is faulty. If policy 2 is worse than policy 1 (suppose it is trained with less data), then would stitching a policy 2 observation to policy 1 decoder result in improved performance? This would be of interest for example if policy 1 is some pretrained open model available, and policy 2 can be cheaply trained in a new setting with a different camera configuration. It would be very useful if the authors can provide performance of the base policy 1 and policy 2 in all their experiments.”**
>
> In our experiments, we trained both policy 1 and policy 2 to high success rates with converged performance in their environments (E_1 and E_2) separately, and then performed perception stitching and deployed the reassembled policy in a new environment (E_3) with a new combination of visual configurations.

---

> > ### Author Response · Authors · 2024-09-30
> > **Response to Reviewer p1mS (part 4)**
> >
> > - **Door**
> >
> > | Method                      | Mask         | Zoom in      | Blurred        | Noise          | Fisheye        | Position       | Lighting       | Average |
> > |-----------------------------|--------------|--------------|----------------|----------------|----------------|----------------|----------------|---------|
> > | **Policy 1**                | 95.3±1.89    | 0.0±0.00     | 19.3±4.71      | 66.0±4.32      | 4.0±1.63       | 0.0±0.00       | 0.7±0.94       | 26.5    |
> > | **Policy 2**                | 89.3±0.94    | 5.3±1.89     | 21.3±5.25      | 52.7±0.94      | 4.0±1.63       | 0.0±0.00       | 14.7±3.40      | 26.8    |
> > | **RT1**                     | **96.0±1.63** | 0.0±0.00     | 26.7±3.40      | **94.7±3.40**  | 0.7±0.94       | 0.0±0.00       | 0.0±0.00       | 31.15   |
> > | **Devin et al. 2017**       | 9.3±4.11     | 5.3±0.94     | 0.0±0.00       | 4.0±1.63       | 0.7±0.94       | 0.0±0.00       | 6.0±1.63       | 3.6     |
> > | **Cannistraci et al. 2024 (linear)**    | 0.0±0.00    | 1.3±0.94     | 10.7±2.49      | 10.7±4.99      | 2.0±1.63       | 47.3±9.29      | 12.0±7.12      | 12      |
> > | **Cannistraci et al. 2024 (non-linear)**| 26.0±2.83    | 31.3±4.99    | 49.3±8.22      | 48.0±5.89      | 62.7±3.40      | 44.7±3.40      | 33.3±4.99      | 42.2    |
> > | **PeS (w/o disent. loss)** | 24.7±7.71    | 44.0±2.83    | 34.7±3.77      | 0.7±0.94       | 36.7±0.94      | 23.3±3.40      | 24.0±2.83      | 26.9    |
> > | **PeS (w. l1 & l2 loss)**   | 4.0±1.63     | **78.0±5.66** | 3.3±0.94      | 2.0±1.63       | 42.7±4.99      | 6.0±3.26       | 14.7±4.99      | 21.5    |
> > | **PeS**                     | 58.7±4.11    | 68.7±0.94    | **70.7±0.94**  | 52.7±3.40      | **64.7±4.99**  | **48.7±3.40**  | **56.7±5.73**  | **60.1** |
> >
> > - **“I dont fully understand the single camera view experiment. If you train the two policies such that one of the encoders contains no information because of the completely dark input, won’t the decoder learn to completely just ignore the latents from that encoder? So for zero shot stitching where you use the policy 1 decoder, wouldn't you expect to get essentially the same performance as policy 1 before stitching? What was the performance of the policies before and after stitching? I would appreciate this section having a bit more clarification.”**
> >
> > Thanks for this thoughtful question, and we would like to clarify this.
> >
> > Ideally, the decoder should learn to completely ignore the latent representations from the encoder with empty input if we train the policy with a large amount of expert data and many epochs. In practice, however, the policy is trained with only 200 expert demonstrations of data and limited epochs. The policy will learn to assign more importance to the encoder with RGB image input and less importance to the encoder with empty input, but it will not fully ignore that encoder with empty input. Therefore, stitching another encoder to replace the encoder with empty input will have less influence on the policy performance compared with the double camera experiments, but the disturbance caused by this new encoder will still exist due to the latent space misalignment, although in some easier experiments the disturbance is marginal. As supported by our single camera experiment results, the performance decreases of the baseline methods after the zero-shot transfer are smaller, compared with that in the double camera experiments, but the performance decreases are not zero.
> >
> > We used this experiment to empirically find that PeS can better alleviate this disturbance of the newly stitched vision encoder compared to other baselines and achieve higher success rates, especially on harder tasks. We believe that this is because PeS can enforce a better latent space alignment.
> >
> > The single camera view experiment section makes our experiment more complete, and we will add the extra clarification of this experiment in the revised version of this paper.

---

> ### Author Response · Authors · 2024-09-30
> **Response to Reviewer p1mS (part 2)**
>
> We agree that reporting the performance of policies 1 and 2 to be stitched in the first place in the new environment (E_3) is a useful baseline. We are happy to report them in the table below and we will add them in our Appendix. We find that the base policies can perform well in some moderate visual variation such as “Mask” and “Noise”, but it has poor performance on some harder tasks with drastic visual configuration changes, such as “Can-Position” and “Stack-Fisheye”. In general, PeS can achieve significantly higher average performance compared with the base policies to be stitched directly in all the five tasks.
>
> We believe that it can be an interesting topic to explore the stitching of two police with different performances. For example, policy 1 is a pretrained open model with strong capability while policy 2 is a small model cheaply trained with less data. Since this is our first step to explore representation alignment for perception stitching and transfer, we leave the exploration of this topic in the future. We appreciate this interesting idea and will add it in the future work section in our revised paper.
>
> - **Push**
> | Method      | Mask         | Zoom in      | Blurred      | Noise        | Fisheye      | Position     | Lighting     | Average |
> |-------------|--------------|--------------|--------------|--------------|--------------|--------------|--------------|---------|
> | **Policy 1** | 94.0±2.83    | 88.7±0.94    | 88.0±1.63    | 74.7±5.73    | 69.3±0.94    | 41.3±0.94    | 38.0±1.63    | 70.6    |
> | **Policy 2** | 96.7±0.94    | 86.0±1.63    | 84.7±3.40    | 73.3±2.49    | 68.7±1.88    | 7.3±0.97     | 36.0±2.83    | 64.7    |
> | **RT1**      | 95.3±2.49    | 89.3±7.36    | 86.0±4.32    | 77.3±8.22    | 76.0±8.49    | 45.3±5.73    | 40.0±6.53    | 72.7    |
> | **Devin et al. 2017** | 60.7±10.6   | 8.7±4.99     | 16.7±3.77    | 59.3±6.80    | 29.3±7.36    | 19.3±5.73    | 36.7±3.40  | 33.0    |
> | **Cannistraci et al. 2024 (linear)** | 89.3±4.11    | 94.0±2.83    | 64.7±1.89    | 74.7±6.18    | 74.0±2.83    | 78.7±2.49    | 87.3±0.94  | 80.4    |
> | **Cannistraci et al. 2024 (non-linear)** | 12.7±1.89    | 18.7±4.99    | 42.8±3.27    | 23.3±0.94    | 6.0±4.32     | 5.3±2.49     | 32.7±3.40  | 20.2    |
> | **PeS (w/o disent. loss)** | **100.0±0.00** | 86.0±2.83    | 80.7±9.84    | **100.0±0.00** | **100.0±0.00** | **100.0±0.00** | 86.7±0.94  | 93.3    |
> | **PeS (w. l1 & l2 loss)** | 88.7±4.99    | 95.3±1.89    | 90.0±5.66    | **100.0±0.00** | 93.3±0.94    | 80.7±4.99    | 89.3±0.94  | 91.0    |
> | **PeS**      | **100.0±0.00** | **100.0±0.00** | **95.3±0.94** | **100.0±0.00** | 92.7±2.50    | **100.0±0.00** | **94.0±4.32** | **97.4** |
>
> - **Lift**
> | Method      | Mask         | Zoom in      | Blurred      | Noise        | Fisheye      | Position     | Lighting     | Average |
> |-------------|--------------|--------------|--------------|--------------|--------------|--------------|--------------|---------|
> | **Policy 1** | 86.0±1.63    | 24.7±3.77    | 68.0±4.90    | 88.0±1.63    | 38.7±5.25    | 0.0±0.00     | 15.3±6.18    | 45.8    |
> | **Policy 2** | 82.0±4.32    | 12.0±7.12    | 91.3±3.40 | 83.3±0.94    | 13.3±0.94    | 0.0±0.00     | 44.0±2.83    | 46.6    |
> | **RT1**      | 86.0±2.82    | 0.0±0.00     | 86.7±6.24    | 85.3±4.99    | 14.0±3.27    | 35.0±7.07    | 46.0±4.32    | 50.4    |
> | **Devin et al. 2017** | 0.0±0.00     | 5.3±2.49     | 48.0±5.89    | 9.3±4.11     | 14.7±4.99    | 36.0±1.63    | 18.7±4.99    | 18.9    |
> | **Cannistraci et al. 2024 (linear)** | 72.7±3.77    | 64.0±2.83    | 86.0±4.32    | 68.7±1.88    | **88.7±1.88** | 57.3±2.49    | 56.0±5.89    | 70.5    |
> | **Cannistraci et al. 2024 (non-linear)** | 89.3±2.49    | 36.0±3.27    | 52.7±3.40    | 93.3±2.49    | 16.7±2.49    | 21.3±0.94    | 37.3±2.49    | 49.5    |
> | **PeS (w/o disent. loss)** | 83.3±6.60    | 80.7±5.73    | **93.3±0.94** | 91.3±5.73    | 79.3±2.49    | **93.3±2.49** | 76.7±3.40    | 85.4    |
> | **PeS (w. l1 & l2 loss)** | **97.3±2.49** | 85.3±0.94    | 90.7±0.94 | 86.0±4.32    | 88.0±1.63    | 84.7±3.77    | 64.7±3.40    | 85.2    |
> | **PeS**      | 92.7±2.50    | **94.7±1.89** | 89.3±4.11    | **96.0±1.63** | **88.7±0.94** | 93.0±0.03    | **84.7±6.60** | **91.3** |

---

> ### Author Response · Authors · 2024-09-30
> **Response to Reviewer p1mS (part 3)**
>
> - **Can**
> | Method                      | Mask         | Zoom in        | Blurred        | Noise          | Fisheye        | Position       | Lighting       | Average |
> |-----------------------------|--------------|----------------|----------------|----------------|----------------|----------------|----------------|---------|
> | **Policy 1**                | 76.7±3.40    | 12.7±0.94      | 80.0±2.83      | 83.3±2.49      | 16.0±1.63      | 0.0±0.00       | 0.0±0.00       | 38.4    |
> | **Policy 2**                | 80.0±0.00    | 4.0±1.63       | 78.0±0.00      | 86.7±0.94      | 13.3±4.11      | 0.0±0.00       | 2.0±0.00       | 37.7    |
> | **RT1**                     | **88.3±6.24** | 0.0±0.00       | **86.7±8.50**  | **88.3±2.36**  | 0.0±0.00        | 8.3±6.24       | 4.0±0.00       | 39.4    |
> | **Devin et al. 2017**       | 19.3±5.25    | 24.7±1.89      | 2.67±1.89      | 6.0±4.32       | 29.3±3.40      | 1.3±1.89       | 3.3±1.89       | 12.4    |
> | **Cannistraci et al. 2024 (linear)**    | 33.3±0.94    | 48.0±1.63      | 48.7±2.49      | 65.3±0.94      | 26.7±3.77      | 34.7±3.77      | 42.7±0.94      | 42.8    |
> | **Cannistraci et al. 2024 (non-linear)**| 72.7±0.94    | 24.7±2.49      | 37.3±4.99      | 42.7±3.40      | 8.7±1.89       | 39.3±1.89      | 38.7±1.89      | 37.7    |
> | **PeS (w/o disent. loss)** | 44.7±8.06    | **89.3±4.11**  | 34.7±4.11      | 30.7±6.80      | **92.7±2.50**  | 44.7±3.40      | 42.7±0.94      | 54.2    |
> | **PeS (w. l1 & l2 loss)**   | 47.3±0.94    | 58.7±1.88      | 54.0±8.64      | 36.0±7.12      | 58.7±1.88      | 64.7±6.60      | 38.0±4.32      | 51.1    |
> | **PeS**                     | 83.3±5.24    | **89.3±2.49**  | 74.0±2.83      | 78.7±4.11      | 56.0±2.83      | **78.7±2.49**  | **73.3±6.80**  | **76.2** |
>
> - **Stack**
>
> | Method                             | Mask         | Zoom in       | Blurred        | Noise          | Fisheye        | Position       | Lighting       | Average |
> |------------------------------------|--------------|---------------|----------------|----------------|----------------|----------------|----------------|---------|
> | **Policy 1**                       | 66.0±2.83    | 16.0±5.89     | 1.3±0.94       | 65.3±0.94      | 26.7±2.49      | 0.0±0.00       | 15.3±6.18      | 27.2    |
> | **Policy 2**                       | 83.3±2.49    | 0.0±0.00      | 4.0±1.63       | 64.0±0.00      | 2.0±0.00       | 2.0±1.63       | 24.0±2.83      | 25.6    |
> | **RT1**                            | 85.0±4.08    | 0.0±0.00      | 0.0±0.00       | 76.7±4.71      | 13.3±5.73      | 0.0±0.00       | 31.3±0.94      | 29.5    |
> | **Devin et al. 2017**              | 0.7±0.94     | 8.0±1.63      | 0.7±0.94       | 24.0±2.83      | 0.0±0.00       | 14.0±3.27      | 4.7±2.49       | 7.4     |
> | **Cannistraci et al. 2024 (linear)**      | 47.3±0.94    | 62.0±4.32     | 32.7±3.77      | 30.7±0.94      | 54.0±8.64      | 14.7±6.18      | 0.7±0.94       | 34.6    |
> | **Cannistraci et al. 2024 (non-linear)**  | 10.0±1.63    | 12.0±0.00     | 0.0±0.00       | 3.3±0.94       | 0.0±0.00       | 0.7±0.94       | 8.7±3.77       | 5.0     |
> | **PeS (w/o disent. loss)**         | 34.0±11.43   | 10.7±4.11     | 62.0±10.71     | 34.0±7.12      | 22.7±3.77      | 26.0±4.32      | 36.7±8.06      | 32.3    |
> | **PeS (w. l1 & l2 loss)**           | 92.7±0.94    | **98.0±0.00** | 62.7±6.60      | 24.0±4.90      | 59.3±7.36      | 58.7±1.88      | 48.0±4.32      | 63.3    |
> | **PeS**                            | **94.7±0.94** | 96.7±0.94     | **90.0±1.63**  | **96.7±1.89**  | **97.3±2.49**  | **80.0±4.90**  | **82.0±1.63**  | **91.1** |

---

> ### Author Response · Authors · 2024-09-30
> **Response to Reviewer p1mS (part 5)**
>
> - **“Given that the main contribution of this work is dependent on this alignment of latents from different views after the relative representation step, it would be nice if latent visualization from other tasks could also be added to the main paper or at least the appendix.”**
>
> We are glad to report that we have visualized the latent representations of other tasks, and the anonymous results can be found in this link: https://sites.google.com/view/pes-latent-representations
>
> In these visualizations, the red data points are the first 5 steps of images in each of the 200 games in the dataset of a certain task. The blue data points are the last 5 steps of images, and the green data points are the 5 steps of images in the middle of each trajectory. Therefore, we have 1000 data points (5 times 200) for each color which represent the starting, middle, and ending stages of a task.
>
> The visualization results show that PeS can better align the latent space and force an approximate invariance of the latent representations. In contrast, the plain baseline (Devin et al. 2017) without adopting the relative representation and the disentanglement regularization leads to different latent representations between the two encoders, and they have an approximately isometric transformation relationship. These visualizations support our conclusions in the paper.

---

> > ### Comment · Reviewer_p1mS · 2024-10-10
> > **Satisfactory response, leaning towards acceptance**
> >
> > Thank you for the detailed response to my concerns. I still think the need to replicate trajectories in the new environment is not as easy as the authors make it out to be in general, even if It might be so for the experimental configurations they have tried in this paper. However, apart from this limitation I don't have any other major concerns. I appreciate the addition of the new experiments by the authors, and improvement (even if slightly) from the base policies is very interesting to see. I do think training with a weak base policy which authors leave for future work would have been a strong addition to the paper, but the current state of the paper is still strong enough for me to lean towards acceptance.

---

> > > ### Author Response · Authors · 2024-10-10
> > > **Thank you**
> > >
> > > We thank the reviewer for the positive feedback and acknowledgment of our contributions. Thank you!

---

### Decision · Action_Editor_WEHU · 2024-11-04

**Recommendation:** Accept as is

**Comment:**

Overall, reviewers are positive about this work and its extensive empirical work. They all agree that the method is novel, interesting, well-supported, and motivates further research. While the reviewers are concerned that the work is fairly narrow, requires possibly specific kinds of data, and is more of a robotics contribution than a fundamental ML contribution, they agree that the method should be of interest to many in the field.

**Audience:**

While the application of the paper's methods is in a somewhat niche area of machine learning around real-world RL and robotics, people in this field would find value in this paper, and its content is relevant to the broader community of RL.

**Claims And Evidence:**

The paper contains a large number of experiments that appropriately support the authors' claims.